# BioCAP: Exploiting Synthetic Captions Beyond Labels in Biological Foundation Models

**Ziheng Zhang**♠† **Xinyue Ma**♠ **Arpita Chowdhury**♠ **Elizabeth G Campolongo**♠
**Matthew J Thompson**♠ **Net Zhang**♠ **Samuel Stevens**♠ **Hilmar Lapp**♣
**Tanya Berger-Wolf**♠ **Yu Su**♠ **Wei-Lun Chao**♠◇ **Jianyang Gu**♠†
♠The Ohio State University  ♣Duke University  ◇Boston University
†{zhang.13617, gu.1220}@osu.edu

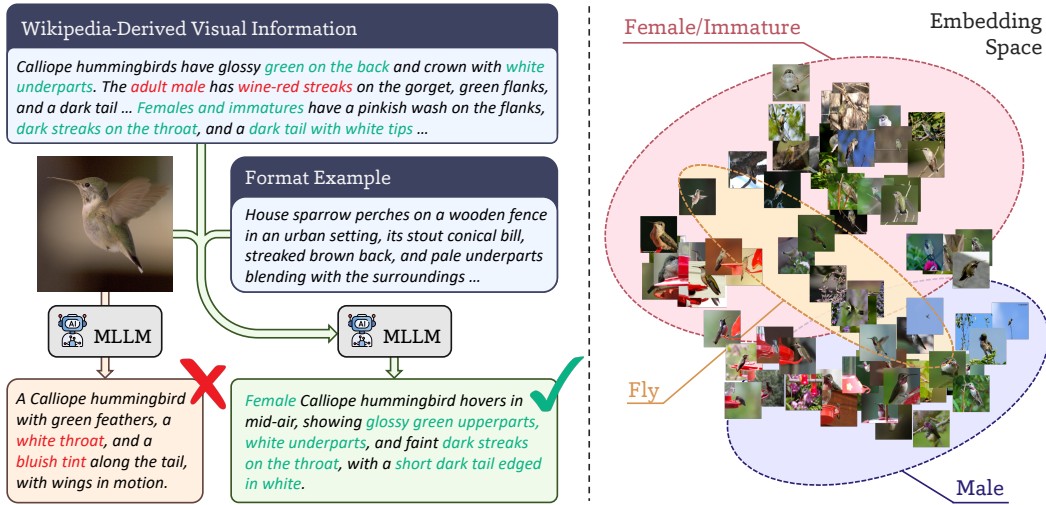

Figure 1: **Left**: **Different strategies to create captions for biological images.** Wikipedia offers rich domain knowledge, but descriptions are often generic and not directly grounded in the given image. Multimodal large language models (MLLMs) may hallucinate details when conditioned solely by images (wrong color description in this example). Incorporating Wikipedia-derived visual information and taxon-tailored format examples as contexts helps generate accurate, image-specific captions. **Right**: Using these descriptive captions as additional supervision, BioCAP captures fine-grained biological semantics. Please refer to Figure 7 for detailed comparisons with other models.

## Abstract

This work investigates descriptive captions as an additional source of supervision for biological multimodal foundation models. Images and captions can be viewed as complementary samples from the latent morphospace of a species, each capturing certain biological traits. Incorporating captions during training encourages alignment with this shared latent structure, emphasizing potentially diagnostic characters while suppressing spurious correlations. The main challenge, however, lies in obtaining faithful, instance-specific captions at scale. This requirement has limited the utilization of natural language supervision in organismal biology compared with many other scientific domains. We complement this gap by generating synthetic captions with multimodal large language models (MLLMs), guided by Wikipedia-derived visual information and taxon-tailored format examples. These domain-specific contexts help reduce hallucination and yield accurate, instance-based descriptive captions. Using these captions, we train BioCAP (*i.e.*, BioCLIP with Captions), a biological foundation model that captures rich semantics and achieves strong performance in species classification and text-image retrieval. These results demonstrate the value of descriptive captions beyond labels in bridging biological images with multimodal foundation models[1].

---

[1]Models, data, and code available at imageomics.github.io/biocap.

# 1 INTRODUCTION

Multimodal foundation models learn from vast datasets of paired visual and textual inputs (Radford et al., 2021; Liu et al., 2023; Zhai et al., 2023; Hurst et al., 2024). They demonstrate strong generalization across various downstream tasks, such as classification and visual question answering (VQA) (Yue et al., 2024; Mai et al., 2025). In general domains, the web provides a nearly limitless supply of such paired supervision (Gadre et al., 2023; Schuhmann et al., 2022). In scientific applications, however, such resources are unevenly distributed. Biomedical imaging benefits from clinical reports and radiology notes associated with image instances that offer detailed natural language supervision (Wang et al., 2022; Li et al., 2023; Zhang et al., 2024a). By contrast, many other scientific domains—including organismal biology, astronomy, geology, and material science (Stevens et al., 2024; Parker et al., 2024; Vivanco Cepeda et al., 2023; Harnik et al., 2025)—often lack instance-level textual descriptions and must solely rely on symbolic labels (*e.g.*, species names). This leaves the potential of natural-language captions untapped for scientific multimodal learning.

In this work, we focus on organismal biology. From the representation learning perspective, each species can be described by an underlying latent vector in the morphospace, which encodes its ground-truth biological characteristics, *i.e.*, traits (Budd, 2021). Images and descriptive captions can be viewed as two projections of this latent vector. Each captures certain traits while also introducing noise (*e.g.*, pose, lighting). Aligning both modalities encourages the model to recover the shared latent structure and focus on potentially diagnostic characters, thereby suppressing reliance on spurious correlations to noise. While the potential exists, in practice, the effectiveness relies on the availability of captions that are both instance-specific and faithful. Noisy or hallucinated captions can, instead, introduce contradictory signals that harm multimodal alignment (Huang et al., 2021).

Biological research has produced massive repositories of organismal images, but most are annotated only with species names (Stevens et al., 2023b; Yang et al., 2024). Collecting instance-based captions at scale is inherently challenging as expert-level annotation for millions of images demands exhaustive human labor (Van Horn et al., 2018). A natural solution is to leverage recent progress in multimodal large language models (MLLMs) to automatically generate image-aligned captions (Fan et al., 2023; Lai et al., 2024). However, biological species differentiation often depends on subtle morphological details. Without proper guidance, MLLMs tend to hallucinate about these details due to the aforementioned noise in the images. Figure 1 shows such an example, where the model incorrectly describes the color of the Calliope hummingbird.

Upon this observation, we suggest that domain knowledge has to be incorporated in the context. Specifically, we collect species-level visual information from Wikipedia (Wikipedia contributors, 2025) as ground-truth characters. As different species feature distinct traits, we also curate format examples based on taxonomic classes to encourage explicit focus. These domain-specific contexts help MLLMs generate accurate and trait-focused descriptions grounded in the input image. As shown in Figure 1, contextualizing the model with Wikipedia-derived visual information and taxon-tailored format examples corrects the hallucination and grounds the caption in accurate biological details.

Building on these curated contexts, we generate instance-level synthetic captions for the large-scale TreeOfLife-10M dataset (Stevens et al., 2023b; Van Horn & Mac Aodha, 2021; Gharaee et al., 2023; Encyclopedia of Life, 2023). We then train the BIOCAP model (*i.e.*, BIOCLIP with Captions) with species names and captions as complementary supervision. As shown on the right of Figure 1, BIOCAP demonstrates a rich understanding of biological semantics. Further comparisons with BIOCLIP (trained without captions) and CLIP in Figure 7 show improved semantic alignment of BIOCAP. Quantitatively, BIOCAP is evaluated on species classification and biological natural-language benchmarks, where it outperforms BIOCLIP by $8.8\%$ and $21.3\%$, respectively. These results answer the core question of this work: descriptive captions, when grounded in biological knowledge, provide essential supervision that bridges organismal images with multimodal understanding.

# 2 RELATED WORK

## 2.1 MULTIMODAL FOUNDATION MODELS FOR SCIENTIFIC IMAGES

Multimodal large language models (MLLMs) such as GPT-4o (Hurst et al., 2024) and LLaVA (Liu et al., 2023) have demonstrated a powerful ability in tasks involving images and natural language

like captioning, VQA, and open-ended reasoning. This paradigm has been successfully extended to specialized domains: LLaVA-Med (Li et al., 2023) adapts LLaVA to radiology for domain-specific VQA, ChemVLM (Li et al., 2025) reasons about molecular structures, and BiomedGPT (Zhang et al., 2024a) unifies diverse biomedical tasks under a single multimodal backbone.

In parallel, CLIP-style contrastive frameworks have also been adapted to a variety of scientific imagery domains. BIOCLIP (Stevens et al., 2024; Gu et al., 2025) and BioTrove-CLIP (Yang et al., 2024) align organismal images with taxonomic names. CLIBD (Gong et al., 2025) uses DNA barcodes as supervision for biodiversity tasks. MedCLIP (Wang et al., 2022) extracts standardized UMLS entities from radiology reports, and MoleCLIP (Harnik et al., 2025) clusters molecular fingerprints in chemistry. CLIP allows for more flexible supervision signals when natural language resources are not available for the specific data. Yet, it also leaves the value of natural language descriptions untapped for these scientific domains. We provide a detailed discussion with concurrent work in §F.

## 2.2 SYNTHETIC CAPTION

One promising solution to overcome the lack of instance-level annotations is to generate synthetic captions that provide fine-grained, image-grounded supervision. Recent work highlights the value of high-quality text-image pairs for CLIP-style training. BLIP (Li et al., 2022) and LLaVA (Liu et al., 2023) are widely used to produce semantically richer descriptions from large-scale web text-image collections. Building on this idea, FG-CLIP (Xie et al., 2025) produces long captions and constructs region-specific annotations, enabling fine-grained alignment. The caption's semantic relevance and granularity are critical for modality alignment. VeCLIP (Lai et al., 2024) and CapsFusion (Yu et al., 2024) refine captions with large language models to achieve more semantically aligned supervision. ALIP (Yang et al., 2023) adopts a bi-path strategy that combines web text with synthetic captions. LaCLIP (Fan et al., 2023) leverages LLMs to generate multiple paraphrases of each caption. These approaches highlight the utility of synthetic captions for improving modality alignment.

Beyond pretraining, synthetic captions have also been applied to specialized settings. Hyp-OW (Doan et al., 2024) employs dense region-level captions to expand visual vocabulary and improve open-world detection. In the MLLM space, MiniGPT-4 (Zhu et al., 2024) and LLaVA rely on GPT-4-generated image descriptions and instruction-following data to equip models with conversational capabilities.

Despite these advances, most existing efforts target general-domain imagery and emphasize caption quality or diversity. Organismal biology introduces additional challenges of fine-grained categorization and domain-specific fidelity. To address these issues, we incorporate domain knowledge into the generation process to reduce hallucination and produce faithful, instance-specific captions.

## 3 METHOD

There have been vast curations of organismal biology images through the efforts of biodiversity researchers and citizen scientists. Many of them have reliable species labels, geolocation metadata, and in some cases, DNA barcodes. However, descriptive captions are largely absent in these datasets. Such captions, which describe human-interpretable visual traits and ecological context, provide complementary information that species labels alone cannot capture. Yet at the same time, their usefulness is dependent on being faithful and instance-specific. Such requirements make them difficult to collect at scale, leaving much of their potential untapped. In this work, we investigate descriptive captions as an additional source of training supervision for biological multimodal foundation models. We first illustrate that descriptive captions, when grounded in biological knowledge, help align images and their species labels. Given the absence of large-scale curated resources, we then explore the use of MLLMs to generate instance-based synthetic captions for biological images.

### 3.1 BIOCAP

Let $x$ be an image embedding, $y \in \mathcal{Y}$ its taxonomic label, and $c$ the corresponding caption embedding. We train BIOCAP (*i.e.*, BIOCLIP with Captions) with two text views: the taxonomic label and the descriptive caption. Both views are encoded by the text encoder and aligned with the image embedding. Assume an underlying trait latent vector $z^*$ in the morphospace encodes the phenotypic characters of taxon $y$. From a causal generation perspective, the image $x$ and caption $c$ are two noisy

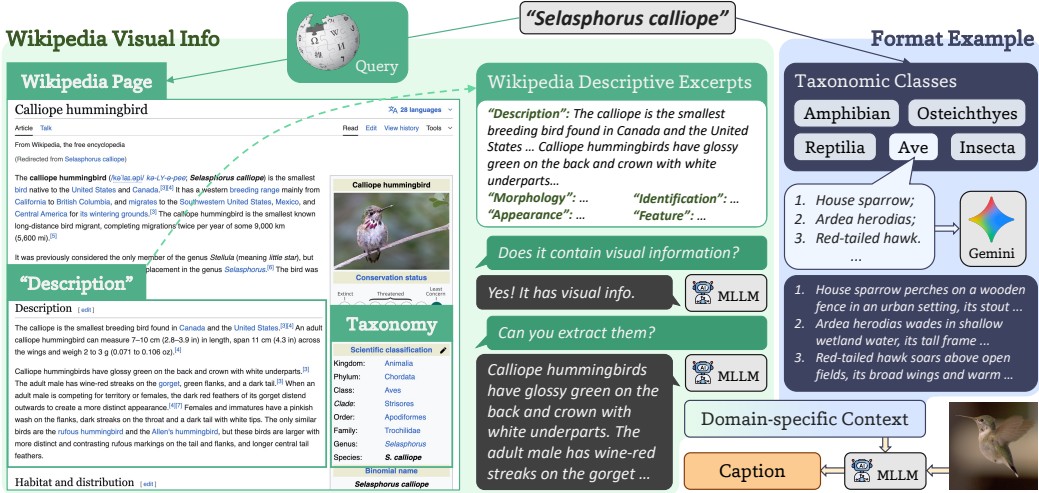

Figure 2: **The pipeline of collecting domain-specific context for MLLMs.** We query Wikipedia with the scientific name to get the corresponding webpage. After validating the full taxonomic rank, we process the descriptive excerpts with MLLMs to extract visual information. For each taxonomic class, we randomly select up to three species and curate format examples through Gemini Deep Research. These contexts help MLLMs generate accurate and grounded descriptive captions.

observations of the same $z^*$:

$$y \rightarrow z^* \rightarrow \{x, c\}, \quad x = g(z^*, \epsilon), \quad c = h(z^*, \epsilon),$$

where $\epsilon$ represents spurious environmental factors (*e.g.*, pose, lighting) that can lead to inaccurate observation. Both the image and the caption capture certain aspects of the ground-truth traits, while being influenced by noise. When captions are involved in the training, the contrastive objective encourages the image embeddings to emphasize trait-relevant factors of the object that are shared with captions, thereby reducing the influence of spurious environmental factors $\epsilon$. Hence, the learned representation becomes closer to the latent trait vector $z^*$ when the caption faithfully reflects visible and potentially diagnostic characters of the species. On the contrary, if the caption captures too much noise instead of the correct traits, the supervision may misguide optimization and degrade classification performance. We further elaborate on this in §A.

**Separated visual projectors.** Given that supervision in our setting is heterogeneous, we introduce two separate visual projectors after the shared visual encoder for taxonomic labels and captions, respectively (Wang et al., 2024; Ranzinger et al., 2024). When the paired text input is a taxonomic label, only the visual embedding after the taxonomy projector is matched, and vice versa. The visual encoder and the text encoder remain shared across both pathways.

## 3.2 SYNTHETIC CAPTION

Based on the above discussion, the captions need to provide thorough and faithful views of the images to assist alignment. Traits that represent the species should be highlighted, while the information not visible or influenced by environmental factors should not be hallucinated. As shown in Figure 1, when solely conditioned by the images, MLLMs tend to generate wrong descriptions due to complicated environmental factors. Therefore, we propose to collect domain-specific contexts to regularize the caption generation. Specifically, we use two context sources: *Wikipedia-derived visual information* and *taxon-tailored format examples*. We present the pipeline for collecting these contexts in Figure 2. The prompts used to collect captions and detailed processes are listed in §B.

**Wikipedia visual information extraction.** Wikipedia provides a comprehensive and accessible knowledge base across the Tree of Life, with species-level descriptions that often contain appearance information. As each instance within a species may have significant appearance variations, these descriptions cannot be directly associated with images. Even so, they offer morphological vocabulary that can be systematically extracted and leveraged in captions.

As shown on the left of Figure 2, we scrape Wikipedia pages based on scientific names and validate the page with the full taxonomy rank. Each page may contain various information regarding the species, such as appearance, habitat, and distribution. We perform a quick filtering to keep the sections that potentially have visual descriptions according to their title (*e.g.*, "Description"). Qwen3 32B (Yang et al., 2025) is then used to verify whether the kept paragraph contains visual information and to extract such information. The model is forced to only focus on attributes such as color, pattern, shape, texture, and other morphological characteristics. When no usable visual information is found for a species, we apply the same process for the corresponding genus and map the description back to the species if available. This pipeline yields 132K descriptions, covering 29.5% of the 447K taxa in TreeOfLife-10M (Stevens et al., 2024). The extracted Wikipedia visual information forms the foundation of the subsequent caption generation process.

**Format example design.** MLLMs might struggle to decide which traits are salient for a given organism. Direct prompting without guidance often leads to hallucination or oversight of important details. Therefore, we design taxon-tailored format examples that illustrate the desired style and content of descriptive captions. For each of the 347 taxonomic classes in TreeOfLife-10M, we query Gemini Deep Research (Comanici et al., 2025) to retrieve candidate textual descriptions of representative species. Each query returns descriptions of up to six species, from which we manually validate trait accuracy and format consistency, and keep at most three per class. When reliable sources are scarce, fewer than three species are included. This process yields 896 curated examples, which explicitly encourage the model to attend to important traits given species labels.

**Caption generation.** With Wikipedia-derived visual information and taxon-tailored format examples as domain-specific contexts, we leverage MLLMs to generate synthetic captions grounded in the target images. Specifically, we use InternVL3 38B (Zhu et al., 2025) as the backbone and accelerate inference with the vLLM framework (Kwon et al., 2023). When no Wikipedia information is available for the species, we only incorporate format examples in the context. Such a process helps the model generate trait-focused descriptions that emphasize visible appearance while avoiding irrelevant or hallucinated details. Furthermore, this pipeline enables large-scale, efficient generation of instance-level captions that complement taxonomy-based supervision.

## 4 EXPERIMENTS

BIOCAP is initialized from the OpenAI ViT-B/16 CLIP checkpoint (Radford et al., 2021) and trained on TreeOfLife-10M (Stevens et al., 2023b; Van Horn & Mac Aodha, 2021; Gharaee et al., 2023; Encyclopedia of Life, 2023) for 50 epochs with species labels and captions. We evaluate BIOCAP on species classification tasks following BIOCLIP 2 (Gu et al., 2025), including NABirds (Van Horn et al., 2015), Meta-Album (Ullah et al., 2022), IDLE-OO Camera Traps (Campolongo et al., 2025; Island-Conservation; Desert-Lion-Conservation, 2024; Vélez et al., 2022; Balasubramaniam, 2024; Yousif et al., 2019), and Rare Species (Stevens et al., 2023a; Encyclopedia of Life, 2023; IUCN, 2022). For evaluating the understanding of natural language, we use INQUIRE-Rerank (Vendrow et al., 2024). In addition, we collect paired text-image data from PlantID (Bruce Homer-Smith and contributors to PlantID.net, 2025) and Cornell Bird (Macaulay Library, Cornell Lab of Ornithology, 2025) to test retrieval performance in organismal domains. Hyperparameter settings and more data information are presented in §C and §D, respectively.

### 4.1 MAIN RESULTS

**Classification.** We evaluate models in the zero-shot setting on ten species classification benchmarks in Table 1. Compared with BIOCLIP, where captions are not involved in training, the accuracy increases by 23.5% on Fungi and 7.1% on Rare Species, demonstrating stronger generalization to challenging real-world settings. Overall, BIOCAP achieves an average top-1 accuracy margin of 8.8% over BIOCLIP and 27.0% over the original CLIP model (Radford et al., 2021).

**Retrieval.** We evaluate the natural language understanding on INQUIRE-Rerank (AP@50), Cornell Bird and PlantID(Recall@10) in Table 2. Except for the Behavior and Context tasks in INQUIRE-Rerank, BIOCAP achieves the best performance across all benchmarks. FG-CLIP is trained for fine-grained retrieval tasks (Xie et al., 2025), yet BIOCAP shows clear advantages with an aver-

Table 1: **Zero-shot species classification top-1 accuracy across 10 tasks for different models.** **Bold** and underlined entries indicate the **best** and second best accuracies, respectively. BIOCAP achieves the best performance across all benchmarks, with an average improvement of $8.8\%$ over BIOCLIP. All the compared models are based on the ViT-B/16 visual encoder.

| | Animals | | | | | Plants & Fungi | | | | | |
| Model | NABirds | Plankton | Insects | Insects 2 | Camera Trap | PlantNet | Fungi | PlantVillage | Med. Leaf | Rare Species | Mean |
|---|---|---|---|---|---|---|---|---|---|---|---|
| Random Guessing | 0.2 | 1.2 | 1.0 | 1.0 | 3.5 | 4.0 | 4.0 | 2.6 | 4.0 | 0.3 | 2.2 |
| CLIP (ViT-B/16) | 39.0 | 3.3 | 7.4 | 9.3 | 28.1 | 52.5 | 8.6 | 5.1 | 15.0 | 25.7 | 19.4 |
| SigLIP | 50.2 | 3.7 | 17.6 | 17.6 | 26.7 | 76.3 | 28.3 | 26.1 | 45.4 | 30.7 | 32.3 |
| FG-CLIP | 48.3 | 1.9 | 6.9 | 26.4 | 26.4 | 55.6 | 7.3 | 5.9 | 15.7 | 29.4 | 20.7 |
| BioTrove-CLIP | 39.4 | 1.0 | 20.5 | 15.7 | 10.7 | 64.4 | 38.2 | 15.7 | 31.6 | 24.6 | 26.2 |
| BIOCLIP | 58.8 | 6.1 | 34.9 | 20.5 | 31.7 | 88.2 | 40.9 | 19.0 | 38.5 | 37.1 | 37.6 |
| BIOCAP | **67.6** | **7.2** | **41.9** | **23.7** | **37.4** | **93.6** | **64.4** | **33.0** | **51.4** | **44.2** | **46.4** |

Table 2: **Performances on natural language tasks, including INQUIRE-Rerank (AP@50) and two text-image retrieval (Recall@10) benchmarks. Bold** and underlined entries indicate the **best** and second best accuracies, respectively. I2T means image-to-text retrieval, and vice versa. With the additional supervision from descriptive captions, BIOCAP achieves an average performance advantage of $21.9\%$ over BIOCLIP.

| | INQUIRE Rerank | | | | Cornell Bird | | PlantID | | |
| Model | Appear. | Behav. | Context | Species | I2T | T2I | I2T | T2I | Mean |
|---|---|---|---|---|---|---|---|---|---|
| CLIP (ViT-B/16) | 30.8 | 32.9 | 37.2 | 37.1 | 33.8 | 33.0 | 26.0 | 23.7 | 31.8 |
| SigLIP | 34.6 | **37.2** | **41.4** | 36.2 | 48.4 | 48.0 | 43.3 | 39.3 | 41.1 |
| FG-CLIP | 28.8 | 31.1 | 32.5 | 41.0 | 50.3 | 48.4 | 28.9 | 28.4 | 36.2 |
| BioTrove-CLIP | 28.5 | 22.2 | 30.5 | 39.5 | 16.6 | 15.3 | 48.0 | 51.5 | 31.5 |
| BIOCLIP | 27.4 | 27.2 | 30.8 | 41.1 | 15.4 | 16.7 | 48.4 | 45.5 | 31.6 |
| BIOCAP | **37.1** | 33.6 | 37.0 | **43.0** | **56.5** | **55.0** | **82.6** | **83.3** | **53.5** |

age performance margin of $17.3\%$. These results indicate that trait-grounded synthetic captions substantially enhance fine-grained, expert-style retrieval of biological knowledge.

## 4.2 ABLATION STUDY

For ablation studies, we train models with TreeOfLife-1M (Stevens et al., 2024) for 100 epochs.

**Influence of different captions.** As illustrated before, the alignment between captions and potential diagnostic traits is critical to the model training. We conduct ablations to examine the impact of different caption generation strategies in Table 3. Using only taxonomic labels without captions (*None*) corresponds to BIOCLIP, and is set as the baseline. Adding raw Wikipedia text as captions is a straightforward way to involve domain knowledge. We demonstrate by *Wiki Page* that using non-instance-specific captions provides improvements in retrieval but does not help classification.

Then we incorporate MLLMs to generate instance-based synthetic captions. The *Base* prompt asks the model to "describe this image in short." Without emphasis on traits, the captions contain inaccurate descriptions and degrade the performance. It corresponds to our previous analysis that noise in captions harms multimodal alignment. The *Trait* prompt explicitly requires the model to focus on traits, which substantially reduces the influence of environmental nuisance. Building upon the *Trait* prompt, we further introduce taxon-tailored format examples and Wikipedia-derived visual information as domain-specific contexts. *Trait+Example+Wiki* demonstrates the best performance, and is used to generate captions for the other experiments. All the adopted prompts are listed in §B.3.

Table 3: **Influence of different captions.** *None*: no caption is used in training (BIOCLIP); *Wiki Page*: a sentence from the Wikipedia visual information; *Synthetic*: captions generated by MLLMs. Synthetic captions involve the option of simple prompts (*Base*) and trait-focused prompts (*Trait*). Domain-specific contexts in MLLM inputs: *Example*: format examples; *Wiki*: Wikipedia-derived visual information. The average performances on each benchmark category (*CLS*: classification; *INQ*: INQUIRE-Rerank) are reported.

| Strategy | | | CLS | Retrieval | | INQ |
| --- | --- | --- | --- | --- | --- | --- |
| Caption | Prompt | Context | | I2T | T2I | |
| None | – | – | 30.2 | 30.5 | 31.2 | 30.0 |
| Wiki Page | – | – | 30.0 | 47.2 | 47.1 | 30.3 |
| Synthetic | Base | – | 27.0 | 26.7 | 28.2 | 29.9 |
| Synthetic | Trait | – | 30.8 | 44.6 | 45.2 | 33.2 |
| Synthetic | Trait | Example | 31.8 | 47.5 | 48.1 | 33.9 |
| Synthetic | Trait | Example+Wiki | **33.8** | **54.7** | **54.3** | **34.8** |

Figure 3: **Ablation study on training recipes.** *200 epochs*: model trained for 200 epochs with species labels; *single proj/dual proj*: projector settings for caption alignment.

Table 4: **The influence of the training set components.** We incrementally add Wikipedia-covered and non-covered species into training with different supervisions to validate the generalization of synthetic captions. The results on the covered/non-covered test species are reported. *Name*: species name; *Caption*: synthetic captions.

Table 5: **Win rate of different synthetic captions in the human evaluation.** *Base*: naive prompt; *Trait*: trait-focused prompt; *Ours*: *Trait*+domain-specific contexts.

| Training Component | | Classification | |
| --- | --- | --- | --- |
| Covered | Non-covered | Covered | Non-covered |
| Name | – | 35.7 | 40.1 |
| Name+Caption | – | 37.5 | 42.7 |
| Name | Name | 35.8 | 43.5 |
| Name+Caption | Name | 38.6 | 46.6 |
| Name+Caption | Name+Caption | **45.3** | **48.8** |

| | Method | | |
| --- | --- | --- | --- |
| Attributes | Base | Trait | Ours |
| Groundedness | 5.7 | 11.9 | **82.4** |
| Specificity | 10.3 | 8.9 | **80.8** |
| Completeness | 5.1 | 7.6 | **87.3** |
| Clarity | 5.1 | 5.7 | **89.2** |
| Average | 6.6 | 8.5 | **85.0** |

**Training recipe.** In BIOCAP, we adopt two separate visual projectors to align species names and captions, respectively. To ablate this design, we train a model with a single visual projector that aligns species and captions at the same time, named as *single proj* in Figure 3. Our two-projector design is named *dual proj*. The results show that *dual proj* consistently outperforms *single proj*, which validates the necessity of decoupling supervision signals in heterogeneous multimodal training. Additionally, caption generation introduces additional computation time. To assess the fairness, we naively double the training epochs and report the results as *200 epochs* in Figure 3. The results show an obvious gap between *200 epochs* and other variants, supporting the worth of synthetic captions.

**Training set components.** Wikipedia provides faithful domain knowledge as contexts for MLLMs. However, it does not cover all the species for usable visual descriptions (See §B.2). We incrementally add different training components to evaluate the generalization of synthetic captions in Table 4. Specifically, we partition the training set based on whether the species are covered by Wikipedia (37.1% of the total taxa in TreeOfLife-1M). We also separate classification benchmarks into the two groups (76.2% covered by Wikipedia). In the table, *Name* refers to taxonomy-only supervision, while *Name+Caption* denotes the joint supervision with synthetic captions. The results show that adding captions to Wiki-covered species also enhances the understanding of non-covered species. Incorporating both taxonomy and captions for all species achieves the best results, even if many species are not covered by Wikipedia, indicating generalization of caption-encoded knowledge.

**Caption Evaluation.** We conduct a human evaluation to assess the quality of the generated captions. As shown in Table 5, we evaluate the captions along four metrics (*Groundedness*: if the description is visible in the image; *Specificity*: if the caption describes distinctive traits; *Completeness*: if the caption covers 2-3 most salient aspects; *Clarity*: preciseness and objectiveness). We randomly sample 200 images from TreeOfLife, covering 200 distinct taxonomic classes. For each image, we provide three captions: one generated by our full method (*Trait+Example+Wiki*) and two obtained from

Table 6: **Influence of example number.** We vary the number of taxon-tailored format examples per class and report the resulting performances.

| Num of Examples | CLS | Retrieval | | INQ |
|---|---|---|---|---|
| | | I2T | T2I | |
| 1 | 33.2 | 52.7 | 53.2 | 33.9 |
| 3 | 33.8 | **54.7** | 54.3 | **34.8** |
| 5 | **34.0** | 54.3 | **54.7** | 34.5 |
| 7 | 33.7 | 54.5 | 54.6 | 34.6 |

Table 7: **Comparison across caption generators and model sizes.** We report the resulting performances under each setting.

| Generator | Size | CLS | Retrieval | | INQ |
|---|---|---|---|---|---|
| | | | I2T | T2I | |
| InternVL3 | 8B | 31.7 | 49.4 | 50.2 | 33.7 |
| InternVL3 | 78B | 33.9 | **54.8** | **55.6** | 34.5 |
| InternVL3 | 38B | 33.8 | 54.7 | 54.3 | 34.8 |
| Qwen-2.5-VL | 32B | **34.1** | 53.3 | 55.5 | **35.2** |
| LLaVA-NeXT | 34B | 32.9 | 54.1 | 54.7 | 34.4 |

| Image & Species | Trait+Example+Wiki (Ours) | Trait | Base |
|---|---|---|---|
| Red Hot Poker | *The red hot poker shows a tall conical inflorescence of orange-to-yellow tubular flowers rising above a pond with water lilies.* | *A striking red hot poker with fiery tubular flowers fading to yellow, rising above pond lilies, attracting hummingbirds nearby.* | *A tall orange and yellow flower standing by the water, shaped like a torch, with green plants around it.* |
| White-Eared Honeyeater | *The White-Eared Honeyeater displays a black head, white ear-coverts, and olive-green body while perched on a branch* | *The White-Eared Honeyeater shows an olive back, pale ear spot, and dark face while standing on a tree.* | *A small bird with green feathers and a white patch sits quietly on a branch.* |
| Common Elbow Orchid | *The common elbow orchid displays small, insect-like flowers on a slender, arching stem with a few basal bracts.* | *The common elbow orchid shows what looks like a small insect clinging to a thin stem with delicate parts.* | *A tiny bug-like shape sits on a stick, seen against a blurry background.* |

Figure 4: **Captions generated by different strategies** (refer to Table 3 and §B.3). The domain-specific contexts of taxon-tailored format examples and Wikipedia-derived visual information significantly reduce hallucination and provide more accurate descriptions of the target object.

alternative strategies (*Base* and *Trait*). The order of captions is randomized. Human evaluators are asked to select the best caption under each metric. Each caption is independently assessed by two evaluators. The results show that captions generated by our full pipeline are recognized by human evaluators to be accurate and image-specific. Detailed statistics are included in §H.

**Ablation on format examples.** Format examples guide MLLMs to emphasize different traits across species. We examine the influence of varying the number of examples per taxonomic class in Table 6. Using only one example for the entire class limits diversity, leading to suboptimal performances. Three or more examples yield similar results. Considering performances and computational cost, we adopt three examples per class in the other experiments. We also include generating format examples at the order level instead of the class level and assess the stability across multiple runs in §E.3.

**MLLM family and model size.** We evaluate using different MLLMs to generate descriptive captions, including LLaVA-NExT (Zhang et al., 2024b), Qwen2.5-VL (Bai et al., 2025), and InternVL3. We also vary the parameter scales of InternVL3 (8B, 38B, 78B) and report the results in Table 7. We observe that captions produced by Qwen2.5-VL and InternVL3 lead to almost the same downstream performance, while LLaVA-generated captions result in slightly lower performance. Overall, the proposed caption generation pipeline is robust across different MLLMs. For model size, the 8B variant produces weaker results, whereas the 38B and 78B models show almost the same performance. Therefore, we select InternVL3 38B for the other experiments. Overall, the caption-generation pipeline does not rely on extremely large or highly specialized MLLMs, and the proposed framework remains stable across both model families and parameter scales.

### 4.3 ANALYSIS AND DISCUSSION

**Qualitative comparison of captions.** Figure 4 illustrates qualitative comparisons of generated captions. The second column corresponds to our full strategy (*Trait+Example+Wiki*). Using the *Base* or *Trait* prompts without domain-specific contexts leads to hallucination (the common elbow

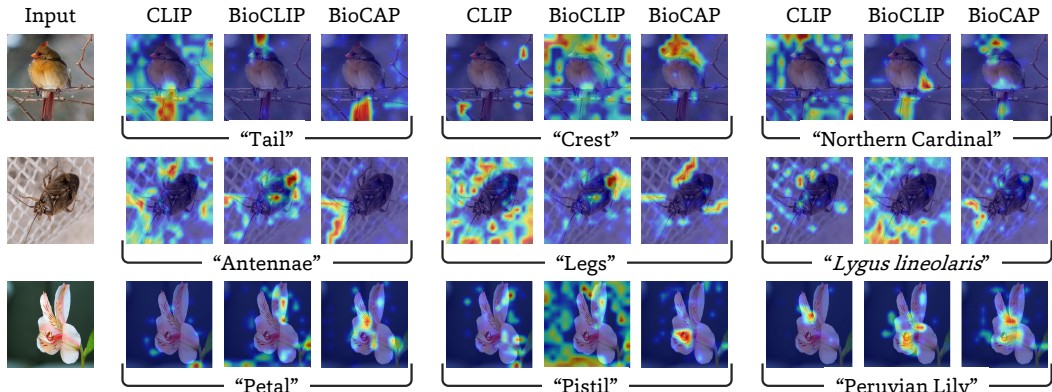

Figure 5: **Grad-CAM visualization of CLIP, BIOCLIP, and BIOCAP**, given *species names* and *biological concepts* frequently mentioned in their captions. Comparatively, BIOCAP offers a comprehensive understanding of these concepts and connects them to the corresponding species.

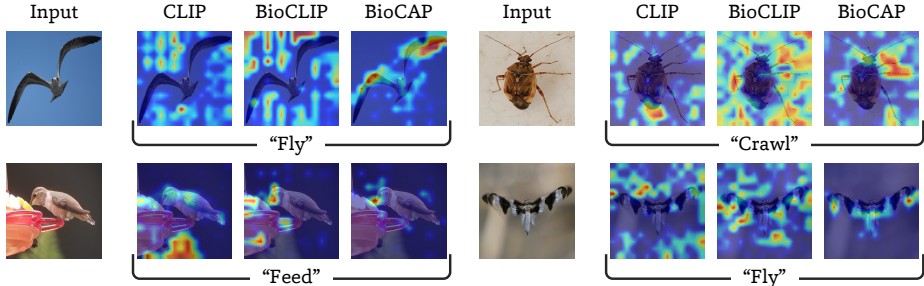

Figure 6: **Grad-CAM visualization of CLIP, BIOCLIP, and BIOCAP**, given *behaviors*. BIOCAP correctly highlights the body parts related to the behaviors.

orchid flower misrecognized as an insect) and inaccurate descriptions (vague color descriptions for the white-eared honeyeater). In contrast, introducing format examples and Wikipedia-derived visual information significantly improves the instance-specificity and faithfulness of the synthetic captions.

**Why are captions helpful for classification?** To better understand the effects of captions, we use Grad-CAM (Zhou et al., 2016) to visualize model attention given species names and high-frequency biological concepts mentioned in the corresponding captions in Figure 5. The visualization shows that BIOCAP learns to localize biologically meaningful traits and associate them with the species names. It corresponds to our analysis in §3.1 that captions help model focus on diagnostic characters during training, and thereby improve classification performance. The information on Grad-CAM implementation and high-frequency concept is presented in §C.3.

**Semantic Understanding.** In addition to specific traits, we further analyze whether captions help align embeddings with broader biologically meaningful semantics (Figure 6). Using Grad-CAM, we visualize model attention given behavior-related words such as fly, feed, and crawl. Compared with CLIP and BIOCLIP, BIOCAP accurately highlights the body parts related to these behaviors, for instance, activation of wings for "fly" and legs for "crawl."

Beyond instance-level understanding, we also visualize the relationship between instances with t-SNE across three bird species in Figure 7, annotated with both behaviors (perch, fly, stand) and sex (male, female/immature). General-purpose models like CLIP and DINOv3 form loose species clusters and conflate sex distinctions. They also mistakenly align female/immature red-winged blackbirds with brown-headed cowbirds. BIOCLIP learns the species distinction, but fails to differentiate the behavior variations. Comparatively, BIOCAP produces compact species clusters and separation of various biological semantics. These results further demonstrate the effectiveness of descriptive captions in enhancing the understanding of various biological concepts. The collection process of behavior labels is presented in §C.4. We provide more qualitative results in §E.

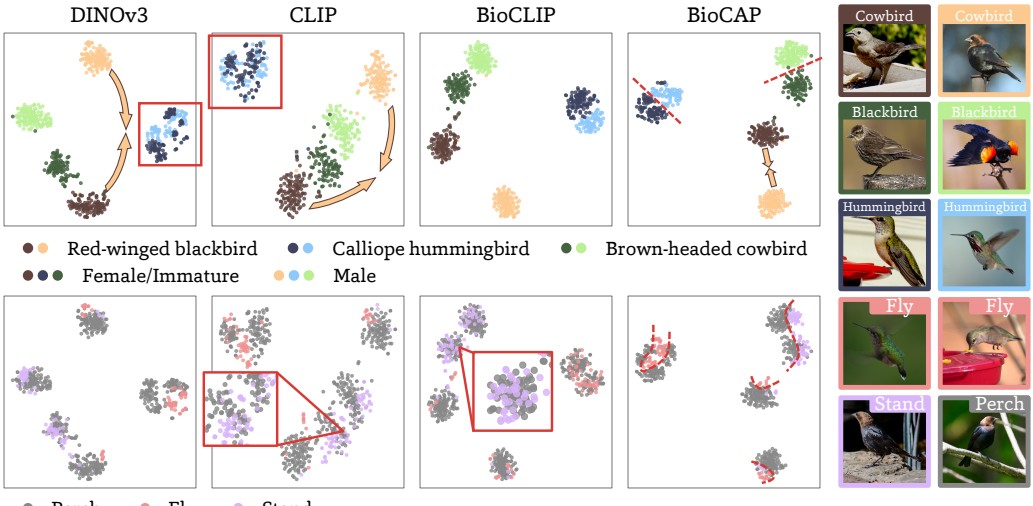

Figure 7: **Embedding distribution of three bird species** with sex and behavior annotations. On the right, we provide example images corresponding to each label. DINOv3 and CLIP fail to align male and female red-winged blackbirds while mixing male and female hummingbirds. BIOCLIP does not capture the semantical difference between behaviors. With the guidance of captions, BIOCAP tells the subtle difference between *perch* and *stand* and accurately separates the behavior variants.

## 5 CONCLUSION

This paper investigates using instance-level descriptive captions as a complementary supervision for biological multimodal foundation models. Due to the lack of such resources at scale, we incorporate multimodal large language models (MLLMs) to generate synthetic captions. We curate Wikipedia-derived visual information and format examples to reduce hallucination and produce accurate, instance-specific captions. Aligning images with captions encourages the model to emphasize potential diagnostic traits while reducing the influence of environmental factors. The acquired BIOCAP model demonstrates rich understanding of a broad range of biological semantics. The superior performance in species classification and biological text-image retrieval highlights the value of descriptive captions in bridging biological images with multimodal foundation models.

## ETHICAL STATEMENT

Our study involves human evaluation of automatically generated captions for organismal biology images. Participants were asked only to express preferences between different caption candidates based on the associated image. No personal or identifiable information was collected, and the task posed minimal risk. Participation was voluntary, and participants were engaged without coercion. As the task did not involve sensitive data, medical decisions, or personal attributes, institutional review board (IRB) approval was not required under our institution's policies. We emphasize that the generated captions are intended for scientific research purposes. Nonetheless, we acknowledge the potential risk of inaccurate or misleading captions, and we therefore recommend their use only as research tools and not as authoritative sources.

## REPRODUCIBILITY STATEMENT

We have released all source code, including modules for model training, evaluation, caption generation, and Wikipedia scraping, at https://github.com/Imageomics/biocap. The corresponding dataset, extended from TreeOfLife-10M with generated captions, is available at TreeOfLife-10M-Captions. The details regarding caption generation are presented in §B. The training hyperparameters and details to reproduce our evaluation results are included in §C. The information on adopted benchmarks is listed in §D.

## ACKNOWLEDGMENTS

We would like to thank Wasila Dahdul, Zhiyuan Tao, Yifan Liu, Fangxun Liu, Shuheng Wang, Ziqi Li, David Carlyn, Quang-Huy Nguyen, Yintie Lei, and Junke Yang for their help with the human evaluation. We also thank Hong-You Chen and the Imageomics Team for their constructive feedback.

We sincerely thank PlantID.net (Bruce Homer-Smith and contributors to PlantID.net, 2025), as well as the Cornell Lab of Ornithology (Macaulay Library, Cornell Lab of Ornithology, 2025) for providing access to their biological media collections. The data made our retrieval evaluation possible.

Our research is supported by NSF OAC 2118240 and resources from the Ohio Supercomputer Center (Center, 1987). The authors are grateful for the generous support of the computational resources from the Ohio Supercomputer Center.

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

# Table of Contents in Appendix

## A  ANALYSIS ON THE IMPACT OF CAPTIONS IN TRAINING

Let $\boldsymbol{x}$ be an image embedding extracted by the visual encoder, $y \in \mathcal{Y}$ the taxonomic label, $\boldsymbol{c}$ the textual embedding of a corresponding descriptive caption, $\boldsymbol{z}^*$ the underlying latent vector of ground-truth traits associated with the taxon $y$, and $\boldsymbol{\epsilon} \perp \boldsymbol{z}^*$ the environmental noise influencing the trait observation. Assume the image $\boldsymbol{x}$ and the caption $\boldsymbol{c}$ are derived from a linear transformation on the latent vector $\boldsymbol{z}^*$ and noise $\boldsymbol{\epsilon}$:

$$\boldsymbol{x} = \boldsymbol{A}\boldsymbol{z}^* + \boldsymbol{G}\boldsymbol{\epsilon} + \eta_x, \quad \boldsymbol{c} = \boldsymbol{B}\boldsymbol{z}^* + \boldsymbol{D}\boldsymbol{\epsilon} + \eta_c,$$

where $\eta_x$ and $\eta_c$ are independent zero-mean Gaussian noises. The label $y$ can be directly determined by $\boldsymbol{z}^*$ and is irrelevant to the noise $\boldsymbol{\epsilon}$. The target is to optimize the encoders so that $\boldsymbol{x}$ is aligned with $\boldsymbol{z}^*$, and thereby the label $y$ can be derived from $\boldsymbol{x}$. We train the model with InfoNCE loss:

$$\mathcal{L}_{\text{NCE}} = \mathbb{E}_{p(\boldsymbol{x},\boldsymbol{c})}\left[-s(\boldsymbol{x},\boldsymbol{c}) + \log \mathbb{E}_{p(\boldsymbol{x}')}\sum \exp s(\boldsymbol{x}',\boldsymbol{c})\right],$$

where

$$s(\boldsymbol{x},\boldsymbol{c}) = \frac{1}{\tau}\phi(\boldsymbol{x})^\top \psi(\boldsymbol{c}).$$

Wang & Isola (2020) show that the contrastive loss optimizes alignment and uniformity properties. The alignment part increases the image-caption inner product $\mathbb{E}[\boldsymbol{x}^\top \boldsymbol{c}]$, while the uniformity part preserves maximum information. For $l_2$-normalized features, $\mathbb{E}[\boldsymbol{x}^\top \boldsymbol{c}] = \text{tr}(\Sigma_{\boldsymbol{x}\boldsymbol{c}})$, *i.e.*, the cross-covariance between the two views. We have:

$$\Sigma_{\boldsymbol{x}\boldsymbol{c}} = \boldsymbol{A}\boldsymbol{B}^\top + \boldsymbol{G}\boldsymbol{D}^\top,$$

which decomposes into the trait-shared component $\boldsymbol{A}\boldsymbol{B}^\top$ and the nuisance-shared component $\boldsymbol{G}\boldsymbol{D}^\top$. If captions capture the diagnostic characters without noise disturbance ($\boldsymbol{D} = 0$), the trait term $\boldsymbol{A}\boldsymbol{B}^\top$ dominates the cross-covariance. The learned image projection, therefore, aligns with trait directions and drops nuisance. Thus, the faithful caption $\boldsymbol{c}$ helps species classification. On the contrary, if the caption covaries with image nuisance $\boldsymbol{\epsilon}$, $\boldsymbol{G}\boldsymbol{D}^\top$ adds to $\Sigma_{\boldsymbol{x}\boldsymbol{c}}$. The InfoNCE loss pulls the image embedding toward the range of $\boldsymbol{G}$, which leads to spurious correlation and potential performance drop. Based on these analyses, we propose to generate synthetic captions that ground biological knowledge to enhance the alignment between images and labels.

# B  DETAILS ON CAPTION GENERATION

In this section, we list the adopted prompts and pipeline details for collecting descriptive captions. We also present some ablation of the visual information extraction pipeline.

## B.1  PROMPTS

**Format example generation.** For each of the 347 taxonomic classes in TreeOfLife-10M, we query Gemini Deep Research (Comanici et al., 2025) to retrieve candidate textual descriptions of representative species. Each query returns up to six candidate descriptions together with images, each representing one species within the class. We manually validate trait accuracy and format consistency based on the images of the species through a winnowing process conducted by two evaluators, both with interdisciplinary training in biology and computer science. Each candidate description is independently evaluated and retained only if it satisfies the following criteria: (1) *visual groundedness*, where all mentioned morphological traits, colors, and structural attributes must be directly observable in the corresponding exemplar image, with additional cross-checking against reliable public sources; (2) *format consistency*, requiring adherence to the predefined caption template with concise descriptions focusing on 2–3 salient visual traits while excluding non-visual attributes such as habitat; and (3) *diversity coverage*, ensuring that the retained examples within each taxonomic class represent distinct species or visually diverse variants rather than redundant descriptions. Candidate descriptions containing hallucinated or unverifiable traits are discarded. The two evaluators first assess candidates independently, and any disagreements are resolved through joint re-examination. We keep at most three curated examples per class, and when reliable sources are scarce, fewer than three species are included. The images are used solely for validation of format examples and are not used in model training. This process yields 896 curated examples, which are then incorporated into the context to guide the model in generating trait-focused captions in a consistent and grounded style. The prompt used for Gemini Deep Research is listed below.

> **Prompt for Format Example Generation**
>
> You are a biologist describing organisms strictly on the basis of visible characteristics in an image. Your task is to generate short, fluent, and biologically meaningful captions for a given class, using examples drawn from different species within that class. Captions must be based on real samples and grounded in visual evidence.
>
> **Requirements:**
> - Provide 6 diverse format examples captions from different species within the class.
> - Captions must emphasize salient visual traits (e.g., color, shape, pattern, texture, body structure).
> - If clearly visible, background or environmental features may be included, but only when they are explicitly apparent in the image.
> - Each caption must contain either the scientific name *or* the common name (not both).
> - Do not begin directly with the name; instead, weave it naturally into the caption text.
> - Each caption must not exceed 35 words.
> - Each caption must be linked to a corresponding image URL, which should point to the actual visual sample used for description.
> - Maintain a concise, scientific style, with variation across examples.
> - If the class contains too few distinctive species, provide fewer than three examples; if no usable information is available, output "N/A".
>
> **Output Format:**
> - Two columns for all provided classes:
>     1. Class name

> 2. Examples (listed as 1, 2, 3, ...), each followed by its corresponding image URL
>
> **Now, generate examples for given classes:** {classes}

**Wikipedia visual information extraction.** Format examples encourage MLLMs to focus on important traits for different species. We also leverage Wikipedia as a large-scale resource to provide detailed visual information across the Tree of Life. The descriptions of visual information on Wikipedia pages are often mixed with habitat, behavior, and distributional information. We perform a quick filtering based on the section titles of the Wikipedia pages and keep those with potential visual information, including "description", "morphology", "appearance", "identification", "feature", "characteristics", "physical", "structure", and "explanation of names."

After the quick filtering, we design a verifying and extraction pipeline to further extract visual information such as color, pattern, shape, and texture. This process yields over 132K trait-focused descriptions, covering 29.5% of the 447K taxa in TreeOfLife-10M (Stevens et al., 2024). The prompt used to filter and extract morphological traits from Wikipedia is provided below.

---

**Prompt for Wikipedia Visual Information Verification**

You are given a textual description of a species.
Your task is to determine whether the description contains any information about the species' **visible appearance** (including features, colors, shapes, patterns, textures, or other morphological characteristics).
Respond strictly with: "Yes" or "No".

**Examples:**

"Bagada is a genus of moths of the family Noctuidae." → No

"Aetheolaena rosana is a species of flowering plant in the family Asteraceae. It is found only in Ecuador. Its natural habitat is subtropical or tropical moist montane forests. It is threatened by habitat loss." → No

"The fur of the African wild dog differs significantly from that of other canids, consisting entirely of stiff bristle-hairs with no underfur. Colour variation is extreme, and may serve in visual identification." → Yes

"The most characteristic physical feature of the raccoon is the area of black fur around the eyes, which contrasts sharply with the surrounding white face coloring." → Yes

**Now classify the following description:**
"{content}"

---

**Prompt for Wikipedia Visual Information Extraction**

You are an expert taxonomy editor. Extract only the sentences (or partial sentences) that describe **visual appearance**:

- Colours, patterns, shapes, sizes, textures, diagnostic marks — anything visible in a photo.
- Visual differences in sex, form, or life stage should be preserved.
- **Do not include** behaviour, distribution, threats, taxonomy, dates, or references.
- Remove all non-visual parts from the original paragraph while maintaining sentence structure.
- Keep exactly the same descriptions from the original input; do not rewrite or rephrase.

Return exactly in the format: <species> | <caption>

---

---

**User Examples:**

> "The fur of the African wild dog differs significantly from that of other canids, consisting entirely of stiff bristle-hairs with no under-fur. Colour pattern is patchy black, yellow ochre and white." → `Lycaon pictus | The fur of the African wild dog consists entirely of stiff bristle-hairs with no under-fur. Colour pattern is patchy black, yellow ochre and white.`

> "The most characteristic physical feature of the raccoon is the area of black fur around the eyes, which contrasts sharply with the surrounding white face colouring." → `Procyon lotor | the area of black fur around the eyes, which contrasts sharply with the surrounding white face colouring.`

> "The male painted bunting is often described as the most beautiful bird in North America... Its colors, dark blue head, green back, red rump, and underparts, make it extremely easy to identify... The plumage of female and juvenile painted buntings is green and yellow-green... The adult female is a brighter, truer green than other similar songbirds." → `Painted Bunting | The male painted bunting has a dark blue head, green back, red rump, and red underparts, making it extremely easy to identify, though it often hides in foliage. The female and juvenile painted buntings have green and yellow-green plumage, which serves as camouflage. The adult female is a brighter, truer green than other similar songbirds.`

**Now extract:**
```
<species>: {species}
<description>: "{description}"
```

---

The design of separating the verification and extraction steps is based on the consideration of efficiency. In our experiment, Qwen3 8B is used for verification and Qwen3 32B is used for extraction (Yang et al., 2025). Compared with a single-step design integrating both verification and extraction with Qwen3 32B, the separate design saves $13\%$ computational time. After manual examination on a 200-sample validation set, the accuracy of Qwen3 8B in verifying if the paragraph contains visual information is consistent with Qwen3 32B. Therefore, we adopt the separate design in our experiment.

**Caption generation.** After acquiring the domain-specific contexts, we query MLLMs to generate instance-based captions. When Wikipedia-derived visual information is not available, only format examples are used in the context. This ensures the model consistently generates accurate and instance-specific descriptions that emphasize visible morphology while avoiding hallucination. The exact prompt template is shown below.

---

Prompt for Caption Generation

You are a biologist describing organisms based strictly on what is visible in the image.
Your goal is to produce a concise caption that highlights diagnostic, image-based traits.
Focus primarily on anatomical structures (e.g., color, shape, pattern, texture, position).
If clearly visible, you may mention substrate, scale cues, or explicit interactions.
Use precise biological terminology. Avoid vague or generic words.

**Examples of good captions:** `{format_examples}`

**If a Wikipedia excerpt is available:**
Reference excerpt about `{species_name}`, use only to standardize correct terms that match visible traits; do not copy text; do not add traits not visible in the image: `{wiki_excerpt}`.

The caption must not exceed `{word_limit}` words.
Include the species name "`{species_name}`" naturally in the sentence.

---

Table 8: Taxa coverage and sample coverage across taxonomic ranks.

| Rank | Taxa coverage | | Sample coverage | |
|---|---|---|---|---|
| | Covered taxa / Total | Ratio | Covered images / Total | Ratio |
| Order | 1,137/1,486 | 76.5% | 9,256,964/9,533,174 | 97.1% |
| Family | 5,127/7,920 | 64.7% | 9,164,272/9,533,174 | 96.1% |
| Genus | 32,725/73,290 | 44.7% | 7,436,080/9,533,174 | 78.0% |
| Species | 122,243/406,293 | 30.0% | 4,989,956/9,533,174 | 52.3% |

> **Priority order:** (1) the most diagnostic visible trait, (2) a secondary distinctive trait, (3) a contextual detail only if it strengthens identification.
>
> **Final instruction:** For the following image of a {`species_name`}, write a single, concise sentence describing its visible traits.

## B.2 STATISTICS OF CAPTION COLLECTION

**Wikipedia coverage.** As illustrated above, we apply an LLM-based extractor to only keep Wikipedia-derived descriptions related to visual information. After excluding non-visual descriptions, we obtain a total of 131,869 Wikipedia-derived visual descriptions, collected through three complementary cases. First, using available species-level labels from TreeOfLife-10M, we collect 109,482 species-level descriptions directly from Wikipedia. Second, for species whose Wikipedia pages lack visual information, we fall back to their corresponding genus pages and map the retrieved genus descriptions back to these species, producing an additional 12,761 species-level descriptions. Together, these two cases provide 122,243 species-level descriptions in total. Third, for samples that lack a species-level label, we use the genus pages directly, adding another 9,626 entries. In total, we obtain 131,869 visual descriptions, consisting of 122,243 species-level and 9,626 genus-level descriptions. These 131,869 descriptions correspond to 29.5% of the 447,593 unique taxa in TreeOfLife-10M.

Table 8 shows the Wikipedia description coverage at different taxonomic levels. One taxon is counted as covered when at least one species within the taxa has associated visual description through the above scraping process. The species-level Wikipedia-derived visual information contributes to 30.0% of the total species-level taxa. 44.7% taxonomic genera have Wikipedia visual information associated with at least one species within them, and the number becomes 76.5% for orders.

Sample-wise, 52.3% samples are covered with the Wikipedia-derived visual information of exactly the corresponding species. When it comes to the family level, 96.1% samples have at least one species covered by Wikipedia within the same family. In such a way, even if the species is not exactly covered, there is at least another similar species with diagnostic characters described in the context. Thereby, the knowledge is able to be generalized across different species during training. We provide quantitative analysis toward the generalization in Table 4.

**Computational time.** For caption generation, we employ InternVL3 38B with the vLLM (Kwon et al., 2023), running on 12 NVIDIA H100 GPUs for about 30 hours to process 10 million samples. Note that the caption generation process is designed to be highly scalable. The adopted vLLM framework allows for efficient distributed inference with optimized memory management and parallel sampling. The system can handle larger datasets by increasing the number of GPUs or extending the running time. This scalability ensures that generating captions for hundreds of millions of images is feasible, which is valuable for existing biological image repositories.

## B.3 BASELINE PROMPTS

In addition to introducing domain-specific contexts, we also explore different formats for the instruction. We design two different prompts: *Base*, which generates generic short captions without any domain hints, and *Trait*, which encourages more detailed, image-grounded trait descriptions. The two prompts are listed below.

---

**Baseline Prompt: Base**

Describe this image in short.

---

**Baseline Prompt: Trait**

You are a biologist describing organisms strictly on the basis of visible characteristics in an image.
Your goal is to produce a concise caption that highlights diagnostic, image-based traits.
Focus primarily on anatomical structures (e.g., color, shape, pattern, texture, position).
If clearly visible, you may mention substrate, scale cues, or explicit interactions.
Use precise biological terminology. Avoid vague or generic words.
The caption must not exceed {`word_limit`} words.
Include the species name "{`species_name`}" naturally in the sentence.

---

We provide the quantitative comparison between the captions generated with the two prompts in Table 3. The qualitative comparisons are presented in Figure 4 and Figure 9. MLLMs tend to produce more detailed descriptions when the prompt explicitly asks the model to focus on traits. Based on the comparison, we use the *Trait* prompt in other experiments and further incorporate domain-specific contexts to ground biological knowledge.

## B.4 GENUS-TO-SPECIES DESCRIPTION MAPPING

| Image & Species | Captions |
|---|---|
| *Moraea flavescens* | *Moraea flavescens features a slender, elongated yellow flower atop a thin green stem, set against a backdrop of dry, woody ground.* |
| *Moraea neglecta* | *Moraea neglecta features bright yellow, iris-like flowers with twisted petals, growing amidst dry, grassy substrate.* |

Figure 8: Examples of genus-level Wikipedia descriptions used as fallback and the resulting captions.

When species-level Wikipedia visual descriptions are unavailable, we map the corresponding genus-level descriptions to species within that genus. This strategy is biologically grounded: in taxonomy, species belonging to the same genus typically share most of their diagnostic morphological characters and differ only in a small subset of traits (Mayr & Ashlock, 1991). Genus-level descriptions therefore provide a coherent approximation of the shared morphological context for closely related species. These descriptions are not used directly as captions, but serve as contextual information to guide the caption-generation MLLM, which still conditions on the input image. As a result, even if multiple species receive the same genus-level Wikipedia visual description, the generation process leads to instance-specific captions that reflect visual traits present in the image, shown in Figure 8.

## C    EXPERIMENTAL DETAILS

We generate synthetic captions using InternVL3 38B. The generation is conducted on 12 NVIDIA H100 GPUs for 30 hours with the vLLM framework (Kwon et al., 2023). We apply nucleus sampling (top-$p = 0.8$) with a temperature of 0.6. With captions and species labels obtained, we train our model on 8 H100 GPUs for 50 epochs using the openclip framework (Cherti et al., 2023). Each GPU processes 4,096 text-image pairs per batch, resulting in a global batch size of 32,768. We use the AdamW optimizer (Kingma & Ba, 2015) with a learning rate of $1 \times 10^{-4}$, weight decay of 0.2, and a linear warm-up during the first 500 iterations. Images are resized to $224 \times 224$ for training and evaluation. As described in §4.2, we adopt a dual-projector design with two separate visual projectors: one for taxonomy supervision and the other for caption supervision. All embeddings used for evaluation are extracted from the taxonomy projector.

### C.1    HYPERPARAMETERS

Table 9: The adopted hyper-parameter setting in training BIOCAP.

| Hyper-parameter | Value |
|---|---|
| Architecture | ViT-B/16 |
| Optimizer | Adam |
| Batch size/GPU (organism) | 4,096 |
| GPUs | 8 H100s |
| Epochs | 50 |
| Max learning rate | $1 \times 10^{-4}$ |
| Warm-up steps | 500 |
| Weight decay | 0.2 |
| Input resolution | 224 |

Table 10: The adopted hyper-parameter setting in ablation study.

| Hyper-parameter | Value |
|---|---|
| Architecture | ViT-B/16 |
| Optimizer | Adam |
| Batch size/GPU (organism) | 4,096 |
| GPUs | 4 H100s |
| Epochs | 100 |
| Max learning rate | $1 \times 10^{-4}$ |
| Warm-up steps | 200 |
| Weight decay | 0.2 |
| Input resolution | 224 |

We summarize the hyper-parameter configurations in Table 9 and Table 10, corresponding to the main training of BIOCAP and the ablation study. The batch size reported in both tables refers to the per-GPU value. Compared with the full training, the ablation uses fewer GPUs and a shorter warm-up schedule, while keeping the overall architecture unchanged.

### C.2    TEXT-IMAGE RETRIEVAL

We evaluate zero-shot retrieval on Cornell Bird  (Macaulay Library, Cornell Lab of Ornithology, 2025), PlantID (Bruce Homer-Smith and contributors to PlantID.net, 2025). We follow the standard text-image retrieval protocol using paired text-image data. Both text-to-image and image-to-text retrieval are considered by embedding the two modalities into a joint representation space and ranking candidates according to cosine similarity. Performance is measured by Recall@10, which reflects how often the correct match is retrieved within the top results.

### C.3    GRAD-CAM VISUALIZATION

**Implementation.** We adopt Grad-CAM (Zhou et al., 2016) for visualizing CLIP attention. Grad-CAM highlights image regions most relevant to a target output by weighting feature maps with their corresponding gradients. For a CLIP model, given an image $\mathbf{I}$ and a text prompt, their embeddings are obtained via the image encoder $f_{\text{img}}$ and text encoder $f_{\text{text}}$:

$$\mathbf{V} = f_{\text{img}}(\mathbf{I}), \qquad \mathbf{T} = f_{\text{text}}(\text{prompt}).$$

The cosine similarity logit is:

$$s = \frac{\mathbf{V} \cdot \mathbf{T}}{\|\mathbf{V}\| \|\mathbf{T}\|}.$$

We backpropagate this logit as the target signal, and apply Grad-CAM to the final transformer block of the image encoder to obtain heatmaps. Since the CLIP image encoder is ViT-based, we reshape the patch-token activations from the target layer into a 2D spatial grid before upsampling.

Table 11: Agreement between GPT-4o, Gemini 2.5 Pro, and human annotators for behavior labels.

| Annotator A | Annotator B | Agreement |
|---|---|---|
| GPT-4o | Human | 95.1 |
| Gemini 2.5 Pro | Human | 92.9 |
| GPT-4o | Gemini 2.5 Pro | 94.0 |

**High-frequency concept collection.** For each target species, we first aggregate all captions and compute the frequency of words after removing common stopwords (*e.g.*, *and*). Then we manually select biologically meaningful concepts starting from the most frequent words, such as body structures (*e.g.*, antennae, petals, tails). The high-frequency concepts are then used as the text prompt for Grad-CAM. This procedure ensures that the concepts used in visualization correspond to the supervision signal of the synthetic captions. Based on the visualization examples, we explicitly show that aligning captions and images guides the model to focus on the diagnostic characters. Thereby, BıoCAP demonstrates better species classification performance.

## C.4 Behavior Semantic Annotation

In addition to static biological concepts related to organs or body parts of the object, we also intend to investigate the model's understanding of behavioral semantics. Given that NABirds (Van Horn et al., 2015) does not provide behavior annotations in the original dataset, we use GPT-4o (Hurst et al., 2024) to automatically assign one of three mutually exclusive behavior categories: *fly*, *perch*, or *stand*. Here, we do not rely on manual labeling, which can be subjective and inconsistent across annotators. These three categories cover the majority of common bird poses. The model is prompted with the following instruction:

> **Prompt for Behavior Annotation**
>
> You are an ornithologist tasked with identifying bird behaviors from images. Looking at this bird image, classify the bird's behavior into exactly ONE of these three categories:
> • `fly` → Wings are spread/extended, bird appears to be in flight or airborne
> • `perch` → Bird's feet are gripping thin branches, reeds, wires, or similar thin supports
> • `stand` → Bird's feet are on ground, soil, rocks, thick surfaces, or flat platforms
> Output only the behavior label: `fly`, `perch`, or `stand`.

To further verify the reliability of these labels, we collect annotations from Gemini 2.5 pro and from human annotators. The agreement rates are summarized in Table 11. Across all comparisons, the agreement exceeds 92%, indicating that behavior classification is a low-ambiguity task. The two MLLMs produce labels highly consistent with human judgment, supporting the robustness of the automatic annotation pipeline.

# D BENCHMARK DETAILS

## D.1 COLLECTION OF RETRIEVAL BENCHMARKS

Table 12: Benchmarks collected for image-text retrieval evaluation.

| Benchmark | Description | Species | Image-Text Pairs |
|---|---|---|---|
| Cornell Bird | Sourced from the Macaulay Library of the Cornell Lab of Ornithology, containing image-text pairs of North American bird species. | 700 | 7,000 |
| PlantID | Collected from PlantID, providing paired images and textual descriptions for a wide range of plants. | 794 | 2,254 |

We collect two retrieval benchmarks to evaluate our model under diverse biological domains: Cornell Bird (Macaulay Library, Cornell Lab of Ornithology, 2025) and PlantID (Bruce Homer-Smith and contributors to PlantID.net, 2025). These datasets cover bird and plant species, each providing paired images and textual descriptions for fine-grained retrieval tasks. Table 12 summarizes the key characteristics of these benchmarks.

## D.2 INFORMATION OF OTHER BENCHMARKS

Table 13: Datasets used for zero-shot classification evaluation. Top-1 accuracy is reported for all the listed benchmarks.

| | Name | | Examples | Classes | Labels |
|---|---|---|---|---|---|
| Animals | NABird | (Van Horn et al., 2015) | 48,000 | 400 | Common |
| | Plankton | (Heidi M. Sosik, 2015) | 4,080 | 102 | Mixed |
| | Insects | (Serret et al., 2019) | 4,680 | 117 | Scientific |
| | Insects 2 | (Wu et al., 2019) | 4,080 | 102 | Mixed |
| Plants & Fungi | PlantNet | (Garcin et al., 2021) | 1,000 | 25 | Scientific |
| | Fungi | (Picek et al., 2022) | 1,000 | 25 | Scientific |
| | PlantVillage | (G. & J., 2019) | 1,520 | 38 | Common |
| | Medicinal Leaf | (S & J, 2020) | 1,040 | 26 | Scientific |
| | PlantDoc | (Singh et al., 2020) | 1,080 | 27 | Common |
| CameraTrap | Desert-lion | (Desert-Lion-Conservation, 2024) | 352 | 32 | Taxonomic |
| | ENA24 | (Yousif et al., 2019) | 1120 | 20 | Taxonomic |
| | Island | (Island-Conservation) | 310 | 17 | Taxonomic |
| | Ohio-small-animals | (Balasubramaniam, 2024) | 468 | 39 | Taxonomic |
| | Orinoquia | (Vélez et al., 2022) | 336 | 28 | Taxonomic |
| | Rare Species | (Stevens et al., 2023a) | 12,000 | 400 | Taxonomic |

We summarize the datasets used for zero-shot classification evaluation in Table 13. These benchmarks cover diverse biological domains, including animals, plants, fungi, and camera-trap datasets, and all tasks are evaluated with Top-1 accuracy. We additionally evaluate on INQUIRE-Rerank (Vendrow et al., 2024), a benchmark where the goal is to re-rank 100 candidate images for each of 200 text queries (20K images in total), ensuring relevant images appear higher in the order.

## D.3 DUPLICATE AND LEAKAGE CONTROL

To ensure rigorous evaluation and eliminate any potential information leakage, we conduct duplicate control for two retrieval benchmarks used in this study. Since the two collected retrieval datasets (PlantID (Bruce Homer-Smith and contributors to PlantID.net, 2025) and Cornell Bird (Macaulay Library, Cornell Lab of Ornithology, 2025)) are uploaded individually by users rather than systematically aggregated, their provenance metadata is noisy and inconsistent. We therefore apply perceptual hashing (Facebook, Inc, 2019) with a distance threshold of 10 to detect visually similar images that cannot be identified through metadata or MD5. Images flagged as near-duplicates are treated conservatively to avoid leakage from any retrieval dataset into the training corpus. Using this pipeline,

we find that $4.1\%$ of PlantID images and less than $0.1\%$ of Cornell Bird images appear in the training set before filtering. All overlapping images are removed. This deduplication ensures that all retrieval benchmarks are cleanly disjoint from the training data.

### D.4 LICENSE INFORMATION

For retrieval evaluation, we additionally used paired text-image data from PlantID (Bruce Homer-Smith and contributors to PlantID.net, 2025) and the Cornell Bird Macaulay Library (Macaulay Library, Cornell Lab of Ornithology, 2025).

PlantID is developed and maintained by Bruce Homer-Smith with contributions from numerous experts and organizations. The website content, including most textual and photographic materials, is released under the Creative Commons CC BY-NC 3.0 license, which allows reuse for non-commercial purposes with proper attribution. We confirm that our usage strictly followed these terms.

The Cornell Bird Macaulay Library is maintained by the Cornell Lab of Ornithology. Use of its media assets in scientific publications requires attribution following their official guidelines. We obtained explicit approval for our use and acknowledge receipt of media from the Cornell Lab of Ornithology | Macaulay Library.

We sincerely thank PlantID, its contributors, and the Cornell Lab of Ornithology for making these invaluable resources available to the community, which enables our retrieval evaluation.

# E MORE QUALITATIVE AND QUANTITATIVE EVALUATIONS

## E.1 SYNTHETIC CAPTION COMPARISONS.

| Image & Species | Trait+Example+Wiki (Ours) | Trait | Base |
|---|---|---|---|
| Lesser Horseshoe-vetch | Lesser Horseshoe-vetch features small, bright yellow flowers with compound, green leaves and reddish stems, growing low to the ground. | The image shows a plant with small yellow flowers and delicate, feathery green leaves, typical of Lesser Horseshoe-vetch, growing among rocks. | Small plant with delicate yellow flowers and green leaves growing in rocky soil. |
| Cloesia digna | Cloesia digna displays iridescent blue wings with a metallic sheen and distinct brownish fringes, set against a plain white background. | The moth has dark, textured wings with a bluish tint and a distinctive body, matching its scientific name Cloesia digna. | A dark-colored moth with intricate patterns on its wings is displayed against a white background. |
| Habronattus fallax | Habronattus fallax exhibits a mottled brown pattern with distinct white stripes on the cephalothorax, positioned on a textured substrate. | The image shows a small spider with a speckled pattern, likely Habronattus fallax, featuring distinct markings and slender legs on a textured surface. | A small spider with striped legs and body sits on a textured surface near a straw. |
| Garden Orbweaver | Garden Orbweavers display a distinctive zigzag stabilimentum in their web, with a central white cross pattern and radial symmetry, set against a backdrop of green foliage. | This is Larvae of the Larvae orb-weaver spider. The intricate web is set in a green, leafy environment, likely a garden or forest. | A spider meticulously creates an intricate, decorative web against a backdrop of green leaves. |
| Paranapiacaba significata | A small, reddish-orange insect, possibly Paranapiacaba significata, is perched on a white, five-petaled flower with dark spots near its center, set against green foliage. | The image shows a white flower with a green center, featuring small black spots and an ant on its petals, set against green foliage. | A white flower with a small insect on it, set against green foliage. |
| Pink Leaf Moth | The Pink Leaf Moth displays vibrant pink and yellow wings with subtle gradation, delicate furry texture, and prominent antennae. | The moth has vibrant pink and yellow wings, a delicate appearance, and a soft, pastel color gradient, typical of the Pink Leaf Moth species. | A colorful moth with pink and yellow wings displayed against a white background. |
| Rattlesnake Mannagrass | Rattlesnake mannagrass displays an open panicle with drooping spikelets, nestled among broad, green leaves in a grassy habitat. sci.txt: a photo of Glyceria canadensis. | The image shows dense, green grass with small, light-colored seed heads, typical of rattlesnake mannagrass, growing in a natural setting. | Close-up of grass with small seed heads in a natural, outdoor setting. |
| Atlantic Threetooth | The Atlantic threetooth features a brown, spiraled shell with fine growth lines and a pale, open umbilicus, resting on a soft, fibrous surface. | The image shows a small, coiled, three-toothed snail with a brown, textured shell resting on a soft, fibrous surface. | A small, coiled snail larva is resting on a soft, fibrous surface. |

Figure 9: More qualitative comparison on the captions generated by different prompts. Our full pipeline, which emphasizes focusing on traits and introduces domain-specific contexts, demonstrates accurate and instance-specific captions.

Figure 9 shows more captions generated by different prompts, including *Base*, *Trait*, and our full pipeline (*Trait+Example+Wiki*). The *Base* captions are often not detailed and not based on biological knowledge. Even if the model is explicitly prompted to focus on traits given the species name, MLLMs can also hallucinate and lose attention to the target object. As shown in the "Paranapiacaba significata" example, the *Trait* caption falsely describes the flower instead of the insect. Our full pipeline, in contrast, accurately describes the insect. When the caption contains too much noise while losing focus on the target object, the provided supervision can misguide the multimodal alignment and harm the model performance.

## E.2 GRAD-CAM RESULTS.

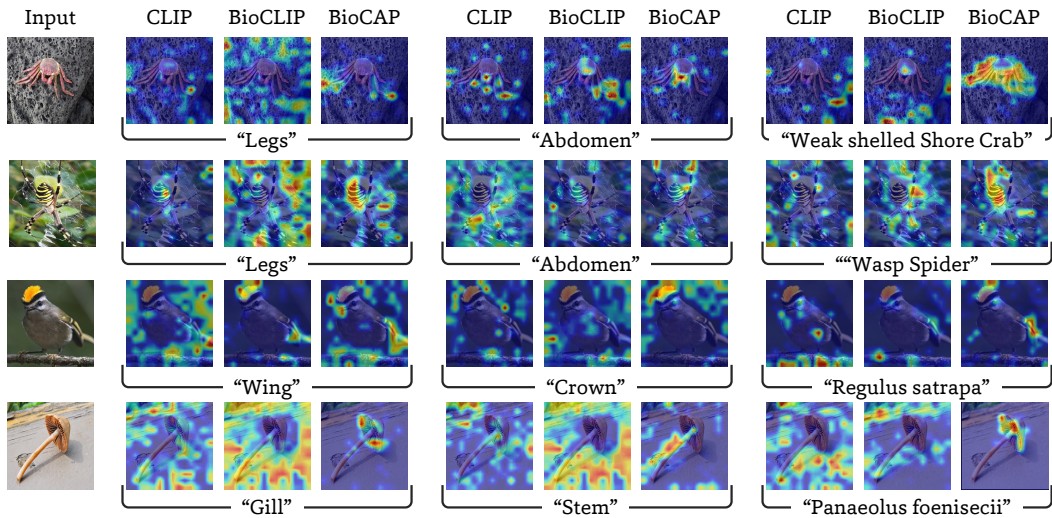

Figure 10: Grad-CAM visualization of CLIP, BIOCLIP, and BIOCAP, given species names and biological concepts frequently mentioned in their captions. BIOCAP accurately highlights these concepts and associate them with classification.

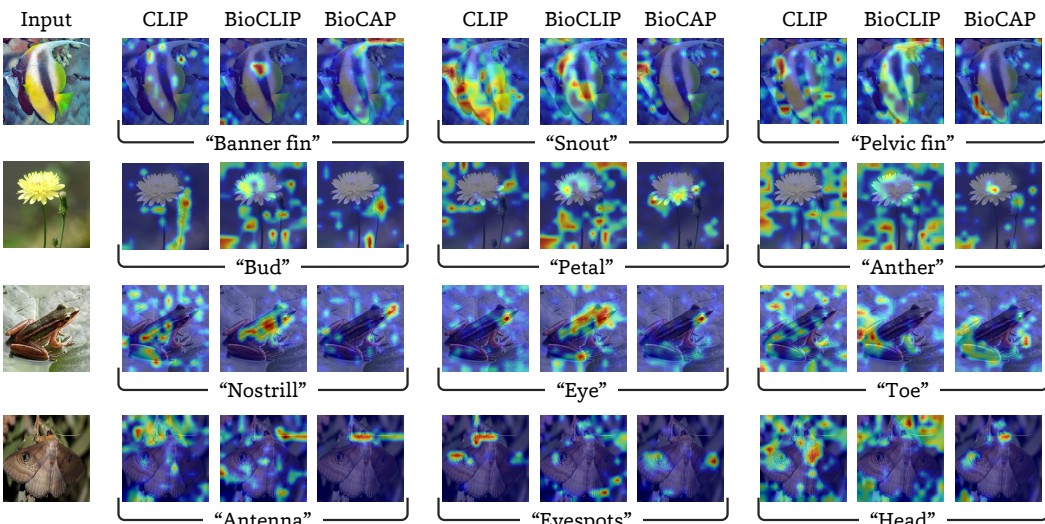

Figure 11: Finer-CAM visualization of CLIP, BIOCLIP, and BIOCAP, given biological concepts frequently mentioned in their captions. BIOCAP accurately localizes these fine-grained concepts.

**Alignment with biological traits.** Various biological concepts are mentioned in the synthetic captions to describe potential diagnostic characters of the species. We present more visualizations regarding these concepts using Grad-CAM and Finer-CAM (Zhang et al., 2025) in Figure 10 and Figure 11 as a supplement to Figure 5. Compared with CLIP and BIOCLIP, BIOCAP demonstrates significantly better localization of these concepts. Moreover, when Grad-CAM is applied to the species name, the parts highlighted by BIOCAP align with these concepts. The results indicate that the synthetic captions guide the model to focus on potential diagnostic traits, while suppressing the spurious correlation. Thereby, BIOCAP demonstrates better species classification performance, which also aligns with our analyses in §3 and §A.

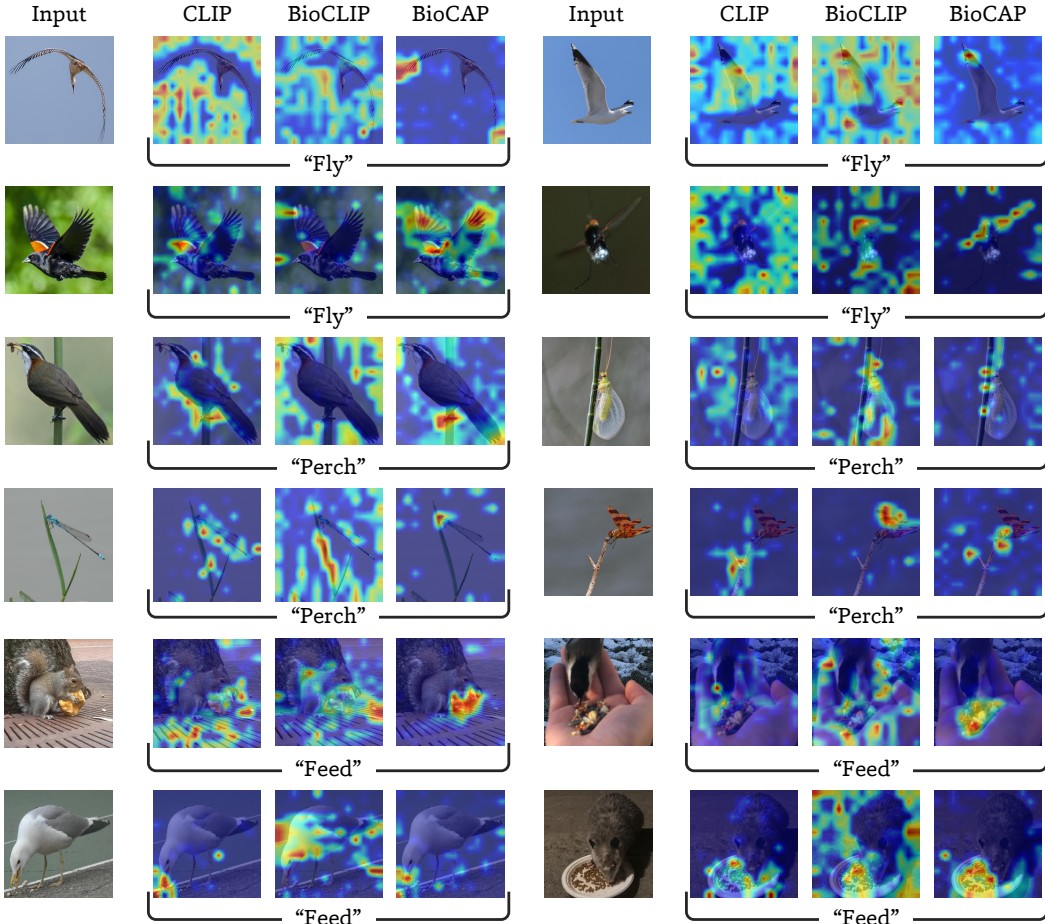

Figure 12: Grad-CAM visualization of CLIP, BioCLIP, and BioCAP, given behaviors. BioCAP correctly highlights the body parts related to the behaviors.

**Alignment with behavior semantics.** In addition to the organism parts that are potentially discriminative traits of the species, the generated captions also contain descriptions regarding behaviors. We provide more Grad-CAM visualizations toward behavior in Figure 12 as a supplement to the previous demonstration in Figure 6. For "fly," "perch," and "feed," BioCAP accurately highlights wings, legs/feet, and mouth/beak/food in the images, respectively. Based on the comparison with CLIP and BioCLIP, the understanding of these behavior concepts is derived from the newly curated synthetic captions. It also validates that our synthetic captions have successfully captured a variety of biological semantics. The rich semantic understanding of BioCAP potentially enables broader applications in various biology-related tasks.

Table 14: Localization performance of different models on CUB, evaluated by the energy-based pointing game.

| Model | Localization score |
|---|---|
| CLIP | 0.36 |
| BioCLIP | 0.43 |
| BioCAP | **0.47** |

**Grad-CAM quantitative results.** We provide a quantitative evaluation of localization quality on CUB (Wah et al., 2011), using ground-truth bounding boxes offered in the dataset. For each model (CLIP, BioCLIP, and BioCAP), we compute Grad-CAM maps with species names as text prompts and measure localization accuracy using the energy-based pointing game (Wang et al., 2020). This

Table 15: **Class-level vs. order-level format examples.** We compare using class and order level examples to guide caption generation.

| Taxonomic Level | CLS | Retrieval | | INQ |
|---|---|---|---|---|
| | | I2T | T2I | |
| Class | **33.8** | 54.7 | **54.3** | **34.8** |
| Order | 33.2 | **55.1** | 53.9 | 33.6 |

Table 16: **Stability across examples generation rounds.** We evaluate BIOCAP using two independently generated exemplar sets.

| Generation Round | CLS | Retrieval | | INQ |
|---|---|---|---|---|
| | | I2T | T2I | |
| 1 | **33.8** | 54.7 | 54.3 | **34.8** |
| 2 | 33.6 | **54.9** | **54.5** | 34.7 |

Table 17: **Performance on underrepresented species groups.** We report classification accuracy across conditions defined by Wikipedia coverage, training image availability, and the Rare Species set. **Bold** and underlined entries indicate the **best** and second best results, respectively.

| Model | Classification | | | | |
|---|---|---|---|---|---|
| | Few-images | | Many-images | | Rare |
| | non-covered | covered | non-covered | covered | |
| CLIP | 16.2 | 32.3 | 21.1 | 23.4 | 25.7 |
| BIOCLIP | 54.0 | 30.4 | 35.4 | 42.1 | 37.6 |
| BIOCAP | **61.0** | **45.0** | **43.7** | **48.9** | **46.4** |

metric quantifies the fraction of activation energy that falls within the annotated bounding box. The results are reported in Table 14. BIOCAP achieves higher localization scores than both CLIP and BioCLIP, indicating that the trait-focused captions help guide the model toward biologically meaningful regions.

### E.3 FORMAT EXAMPLE DESIGN

In §4.2, we analyze the effect of format examples count and show that using three curated examples per taxonomic class provides a stable and sufficient guidance for caption generation. Here, we provide detailed results for two more design choices: the taxonomic level used to generate format examples, and the stability of the pipeline across independent generation rounds.

**Taxonomic level of format examples.** We compare class-level and order-level format examples for guiding caption generation, with results shown in Table 15. The two configurations yield comparable performance across all metrics, while order-level format examples introduce slightly higher variability in a few cases. These findings indicate that class-level format examples strike an effective balance between trait specificity and taxonomic coverage without over-constraining the captioner.

**Stability across generation rounds.** To assess robustness against sampling variability, we regenerate a full second set of format examples using the same Gemini-based pipeline and retrain BIOCAP. As reported in Table 16, while the regenerated format examples are different, the resulting models achieve nearly identical performance. This shows that the format examples generation process remains stable across model runs.

### E.4 PERFORMANCE ON UNDERREPRESENTED SPECIES

It has been shown in Table 4 that BIOCAP improves the species not covered by Wikipedia-derived visual descriptions. We further partition species in the test sets using two criteria: whether the species is Wikipedia-covered or non-covered, and the number of available training images. We rank species by sample count and define the bottom 5% as "few-image" species, with the remainder treated as "many-image." Combining these two factors yields four groups: *few-image + non-covered*, *few-image + covered*, *many-image + non-covered*, and *many-image + covered*. We report the results of these groups in Table 17, together with Rare Species, of which the species are not seen during training. Across all groups, BIOCAP consistently outperforms CLIP and BIOCLIP. The improvements are most pronounced for the two most challenging conditions (*few-image + non-covered* and *few-image + covered*), indicating that caption-guided training is beneficial even when both visual data and external descriptions are limited. BIOCAP also shows clear gains on the Rare Species set, suggesting that

Table 18: **Few-shot species classification top-1 accuracy across 10 tasks. Bold** and underlined values indicate the **best** and second best results. All models use the ViT-B/16 visual encoder.

| | Animals | | | | | Plants & Fungi | | | | | |
| Model | NABirds | Plankton | Insects | Insects 2 | Camera Trap | PlantNet | Fungi | PlantVillage | Med. Leaf | Rare Species | Mean |
|---|---|---|---|---|---|---|---|---|---|---|---|
| *One-Shot Classification* | | | | | | | | | | | |
| CLIP (ViT-B/16) | $24.5_{\pm1.0}$ | $21.8_{\pm1.1}$ | $19.8_{\pm0.5}$ | $11.3_{\pm0.5}$ | $34.0_{\pm2.7}$ | $38.9_{\pm3.7}$ | $15.5_{\pm2.2}$ | $46.0_{\pm2.2}$ | $67.3_{\pm2.4}$ | $26.5_{\pm0.5}$ | 30.6 |
| SigLIP | $30.3_{\pm0.8}$ | $28.2_{\pm0.7}$ | $27.5_{\pm0.9}$ | $17.1_{\pm1.3}$ | $\underline{35.1}_{\pm2.8}$ | $57.1_{\pm4.1}$ | $21.9_{\pm1.8}$ | $58.9_{\pm2.4}$ | $79.6_{\pm1.9}$ | $32.7_{\pm0.3}$ | 38.8 |
| Supervised-IN21K | $45.4_{\pm0.5}$ | $25.6_{\pm0.9}$ | $23.9_{\pm0.9}$ | $20.8_{\pm1.0}$ | $34.3_{\pm2.5}$ | $58.2_{\pm4.4}$ | $28.6_{\pm3.7}$ | $59.5_{\pm1.9}$ | $81.3_{\pm1.8}$ | $36.1_{\pm0.6}$ | 41.4 |
| DINOv3 | $47.8_{\pm0.7}$ | $\mathbf{36.4}_{\pm1.1}$ | $10.1_{\pm0.4}$ | $19.3_{\pm0.6}$ | $\mathbf{43.0}_{\pm2.5}$ | $60.7_{\pm3.5}$ | $23.8_{\pm1.9}$ | $\underline{66.4}_{\pm2.1}$ | $\mathbf{94.3}_{\pm1.5}$ | $41.7_{\pm0.7}$ | 44.4 |
| BioTrove-CLIP | $\mathbf{61.9}_{\pm0.6}$ | $26.4_{\pm0.5}$ | $\mathbf{57.1}_{\pm1.4}$ | $\underline{20.9}_{\pm0.7}$ | $31.2_{\pm2.3}$ | $\mathbf{69.7}_{\pm3.4}$ | $\mathbf{47.3}_{\pm2.1}$ | $55.8_{\pm3.4}$ | $83.5_{\pm1.1}$ | $34.9_{\pm0.4}$ | 48.9 |
| BıoCLIP | $\underline{57.4}_{\pm1.2}$ | $29.7_{\pm1.1}$ | $\mathbf{57.1}_{\pm1.0}$ | $20.4_{\pm0.9}$ | $35.0_{\pm2.8}$ | $67.7_{\pm3.9}$ | $\underline{44.6}_{\pm2.0}$ | $59.5_{\pm2.5}$ | $83.7_{\pm1.8}$ | $\underline{44.9}_{\pm0.7}$ | $\underline{50.0}$ |
| BıoCAP | $53.9_{\pm1.0}$ | $\underline{31.2}_{\pm0.7}$ | $\underline{53.9}_{\pm1.0}$ | $\mathbf{23.5}_{\pm1.3}$ | $33.1_{\pm3.0}$ | $\underline{68.9}_{\pm3.9}$ | $41.2_{\pm1.4}$ | $\mathbf{66.9}_{\pm1.8}$ | $\underline{86.9}_{\pm1.8}$ | $\mathbf{45.4}_{\pm0.8}$ | **50.5** |
| *Five-Shot Classification* | | | | | | | | | | | |
| CLIP (ViT-B/16) | $48.2_{\pm0.3}$ | $36.2_{\pm0.7}$ | $36.7_{\pm0.6}$ | $22.0_{\pm0.1}$ | $51.7_{\pm1.8}$ | $59.6_{\pm2.1}$ | $24.1_{\pm2.1}$ | $69.9_{\pm1.2}$ | $86.1_{\pm0.8}$ | $43.3_{\pm0.3}$ | 47.8 |
| SigLIP | $54.2_{\pm0.4}$ | $47.9_{\pm0.6}$ | $48.0_{\pm0.8}$ | $30.2_{\pm0.7}$ | $52.2_{\pm2.0}$ | $76.6_{\pm1.8}$ | $36.2_{\pm2.0}$ | $78.5_{\pm0.7}$ | $92.4_{\pm1.7}$ | $50.8_{\pm0.4}$ | 56.7 |
| Supervised-IN21K | $66.7_{\pm0.1}$ | $51.0_{\pm0.4}$ | $47.7_{\pm0.6}$ | $35.9_{\pm1.2}$ | $\underline{57.6}_{\pm2.2}$ | $80.7_{\pm1.6}$ | $51.5_{\pm1.6}$ | $83.5_{\pm1.3}$ | $96.5_{\pm1.2}$ | $57.7_{\pm0.2}$ | 62.9 |
| DINOv3 | $75.5_{\pm0.4}$ | $\mathbf{61.0}_{\pm1.0}$ | $28.7_{\pm0.5}$ | $\underline{37.1}_{\pm1.4}$ | $\mathbf{69.3}_{\pm2.2}$ | $\mathbf{86.3}_{\pm1.5}$ | $50.3_{\pm2.0}$ | $\underline{85.6}_{\pm1.7}$ | $\mathbf{99.2}_{\pm0.5}$ | $\mathbf{67.5}_{\pm0.4}$ | 66.1 |
| BioTrove-CLIP | $\mathbf{78.5}_{\pm0.2}$ | $44.6_{\pm0.6}$ | $77.0_{\pm0.6}$ | $34.2_{\pm0.6}$ | $47.9_{\pm2.0}$ | $\underline{86.0}_{\pm1.0}$ | $\mathbf{65.2}_{\pm0.8}$ | $75.1_{\pm0.6}$ | $96.2_{\pm0.7}$ | $51.3_{\pm0.2}$ | 65.6 |
| BıoCLIP | $\underline{78.2}_{\pm0.3}$ | $49.2_{\pm1.1}$ | $\mathbf{78.0}_{\pm0.6}$ | $33.9_{\pm0.6}$ | $54.3_{\pm2.2}$ | $85.7_{\pm1.7}$ | $61.6_{\pm1.9}$ | $81.7_{\pm1.1}$ | $96.7_{\pm0.6}$ | $65.7_{\pm0.4}$ | $\underline{68.5}$ |
| BıoCAP | $77.0_{\pm0.3}$ | $\underline{51.1}_{\pm0.8}$ | $\underline{77.0}_{\pm0.4}$ | $\mathbf{38.1}_{\pm0.4}$ | $48.4_{\pm2.4}$ | $85.4_{\pm1.8}$ | $\underline{63.2}_{\pm3.1}$ | $\mathbf{86.2}_{\pm0.5}$ | $\underline{96.9}_{\pm0.3}$ | $\underline{67.3}_{\pm0.4}$ | **69.1** |

Table 19: **Biological visual tasks beyond species classification. Bold** and underlined entries indicate the **best** and second best accuracies.

| | Animals | | | Plants | | |
| Model | FishNet | NeWT | AwA2 | Herb. 19 | PlantDoc | Mean |
|---|---|---|---|---|---|---|
| CLIP (ViT-B/16) | $25.3_{\pm0.1}$ | $79.7_{\pm0.2}$ | $\underline{66.0}_{\pm0.6}$ | $15.6_{\pm0.2}$ | $17.5_{\pm3.3}$ | 40.8 |
| SigLIP | $\underline{31.9}_{\pm0.1}$ | $83.2_{\pm0.1}$ | $\mathbf{67.3}_{\pm0.6}$ | $18.6_{\pm0.2}$ | $28.2_{\pm5.3}$ | 45.8 |
| Supervised-IN21K | $29.4_{\pm0.1}$ | $75.8_{\pm0.2}$ | $52.7_{\pm1.6}$ | $14.9_{\pm0.1}$ | $25.1_{\pm1.1}$ | 39.6 |
| DINOv3 | $\mathbf{37.9}_{\pm0.1}$ | $\mathbf{85.7}_{\pm0.0}$ | $48.0_{\pm2.8}$ | $\mathbf{31.2}_{\pm0.2}$ | $\mathbf{40.3}_{\pm1.2}$ | $\underline{48.6}$ |
| BioTrove-CLIP | $22.1_{\pm0.0}$ | $82.5_{\pm0.1}$ | $45.7_{\pm0.7}$ | $20.4_{\pm0.2}$ | $37.7_{\pm1.2}$ | 41.7 |
| BıoCLIP | $30.1_{\pm0.2}$ | $82.7_{\pm0.1}$ | $65.9_{\pm0.3}$ | $26.8_{\pm0.4}$ | $\underline{39.5}_{\pm2.3}$ | 49.0 |
| BıoCAP | $29.5_{\pm0.3}$ | $\underline{84.5}_{\pm0.2}$ | $65.6_{\pm1.1}$ | $\underline{28.1}_{\pm0.1}$ | $37.7_{\pm3.1}$ | **49.1** |

the morphological priors encoded in synthetic captions support strong generalization even to species unseen during training.

## E.5 PERFORMANCE ON FEW-SHOT CLASSIFICATION AND BIOLOGICAL VISUAL TASKS

Beyond zero-shot classification and retrieval, we further evaluate BıoCAP on few-shot species classification and additional biological visual tasks (Khan et al., 2023; Van Horn et al., 2021; Xian et al., 2018; Tan & Liu, 2019; Singh et al., 2020) in Table 18 and Table 19, respectively. We observe an overall improvement of BıoCAP over BıoCLIP in few-shot classification and a similar result in other biological visual tasks. We believe the smaller gap compared with zero-shot and multimodal retrieval is expected, given the nature of the added supervision. Compared with species names alone, descriptive captions introduce more "disturbances" that enrich the semantics carried by the embedding but also distract embeddings from the species prototypes. BıoCAP yields a more interpretable intra-class structure tied to biological semantics. However, this supervision does not explicitly enforce more separation between visual embeddings of different species. Therefore, when captions provide better semantic organization and multimodal alignment, they do not contribute to better few-shot classification performances. This is supported by results in Table 20. We compare models trained with varying caption qualities. We observe that while higher-quality captions significantly boost zero-shot and retrieval metrics shown in Table 2, few-shot performance shows minimal variance. This indicates that few-shot capability is weakly correlated with caption quality. On the other hand, the

Table 20: **Effects of different captions on few-shot species classification top-1 accuracy across 10 tasks.** *None* uses no caption; *Wiki Page* uses Wikipedia visual text; *Synthetic* uses MLLM-generated captions via simple (*Base*) or trait-focused (*Trait*) prompts. Domain-specific contexts include format examples (*Example*) and Wikipedia-derived info (*Wiki*). The results demonstrate that varying caption generation strategies yields marginal performance differences in the few-shot setting. **Bold** and underlined values indicate the **best** and second best results.

| | | | Animals | | | | | Plants & Fungi | | | | | |
| --- | --- | --- | --- | --- | --- | --- | --- | --- | --- | --- | --- | --- | --- |
| | **Strategy** | | NABirds | Plankton | Insects | Insects 2 | Camera Trap | PlantNet | Fungi | PlantVillage | Med. Leaf | Rare Species | Mean |
| Caption | Prompt | Context | | | | | | | | | | | *One-Shot Classification* |
| None | - | - | $\mathbf{45.1}_{\pm0.9}$ | $29.9_{\pm1.2}$ | $\mathbf{45.6}_{\pm1.5}$ | $18.7_{\pm0.6}$ | $33.3_{\pm3.6}$ | $\mathbf{64.1}_{\pm3.7}$ | $\mathbf{35.8}_{\pm1.8}$ | $59.8_{\pm2.7}$ | $78.0_{\pm1.7}$ | $36.9_{\pm0.5}$ | $44.7$ |
| Wiki Page | - | | $40.8_{\pm1.0}$ | $\underline{31.9}_{\pm1.7}$ | $42.4_{\pm1.6}$ | $\underline{19.5}_{\pm1.0}$ | $\mathbf{35.3}_{\pm2.5}$ | $60.7_{\pm3.9}$ | $35.0_{\pm0.6}$ | $58.9_{\pm3.4}$ | $\mathbf{82.1}_{\pm1.7}$ | $\mathbf{38.1}_{\pm0.7}$ | $44.5$ |
| Synthetic | Base | | $38.6_{\pm0.8}$ | $\mathbf{34.3}_{\pm0.6}$ | $40.3_{\pm1.5}$ | $\mathbf{20.0}_{\pm0.7}$ | $\underline{34.6}_{\pm3.4}$ | $58.1_{\pm2.2}$ | $34.0_{\pm2.1}$ | $\mathbf{62.2}_{\pm3.9}$ | $\underline{81.9}_{\pm2.7}$ | $35.9_{\pm0.5}$ | $44.0$ |
| Synthetic | Trait | | $41.5_{\pm0.8}$ | $29.9_{\pm1.4}$ | $42.1_{\pm1.2}$ | $\underline{19.5}_{\pm0.9}$ | $34.3_{\pm3.2}$ | $63.1_{\pm3.6}$ | $35.3_{\pm1.8}$ | $60.9_{\pm3.3}$ | $81.1_{\pm2.1}$ | $37.2_{\pm0.8}$ | $44.5$ |
| Synthetic | Trait | Example | $41.9_{\pm0.7}$ | $31.5_{\pm1.3}$ | $\underline{43.5}_{\pm1.4}$ | $19.3_{\pm0.8}$ | $34.5_{\pm2.9}$ | $\mathbf{64.1}_{\pm4.3}$ | $34.9_{\pm1.3}$ | $61.5_{\pm3.1}$ | $79.6_{\pm2.1}$ | $36.8_{\pm0.4}$ | $\underline{44.8}$ |
| Synthetic | Trait | Example+Wiki | $\underline{43.3}_{\pm0.9}$ | $31.8_{\pm1.6}$ | $\underline{43.5}_{\pm0.7}$ | $19.3_{\pm1.0}$ | $33.8_{\pm4.6}$ | $\underline{63.5}_{\pm0.6}$ | $\underline{35.6}_{\pm2.9}$ | $\underline{61.6}_{\pm1.6}$ | $78.7_{\pm0.7}$ | $\underline{37.8}_{\pm0.7}$ | $\mathbf{44.9}$ |
| Caption | Prompt | Context | | | | | | | | | | | *Five-Shot Classification* |
| None | - | - | $67.0_{\pm0.4}$ | $\mathbf{55.7}_{\pm0.6}$ | $65.0_{\pm0.4}$ | $\mathbf{34.8}_{\pm0.6}$ | $52.4_{\pm2.2}$ | $78.3_{\pm1.0}$ | $50.6_{\pm1.1}$ | $\mathbf{84.2}_{\pm0.9}$ | $\mathbf{95.8}_{\pm0.8}$ | $57.1_{\pm0.3}$ | $64.1$ |
| Wiki Page | - | | $\mathbf{69.0}_{\pm0.3}$ | $49.2_{\pm0.7}$ | $\mathbf{69.9}_{\pm0.7}$ | $33.4_{\pm0.6}$ | $\underline{53.5}_{\pm1.2}$ | $\mathbf{82.0}_{\pm1.1}$ | $\underline{54.6}_{\pm1.0}$ | $79.6_{\pm0.7}$ | $93.3_{\pm0.3}$ | $57.9_{\pm0.3}$ | $64.3$ |
| Synthetic | Base | | $66.2_{\pm0.3}$ | $\underline{55.0}_{\pm0.8}$ | $67.3_{\pm0.7}$ | $\underline{34.5}_{\pm0.4}$ | $\mathbf{55.3}_{\pm2.0}$ | $78.9_{\pm1.4}$ | $\mathbf{54.8}_{\pm1.2}$ | $82.5_{\pm1.0}$ | $93.6_{\pm1.0}$ | $58.2_{\pm0.3}$ | $\mathbf{64.6}$ |
| Synthetic | Trait | | $67.0_{\pm0.3}$ | $49.7_{\pm0.5}$ | $67.7_{\pm0.3}$ | $\underline{34.5}_{\pm0.7}$ | $52.7_{\pm2.4}$ | $80.2_{\pm0.6}$ | $53.9_{\pm1.1}$ | $\underline{83.2}_{\pm0.8}$ | $\underline{94.5}_{\pm0.6}$ | $58.2_{\pm0.4}$ | $64.2$ |
| Synthetic | Trait | Example | $67.0_{\pm0.3}$ | $51.6_{\pm0.6}$ | $68.1_{\pm0.6}$ | $33.9_{\pm0.3}$ | $53.3_{\pm2.1}$ | $81.0_{\pm1.0}$ | $54.5_{\pm1.1}$ | $81.8_{\pm1.2}$ | $93.2_{\pm0.6}$ | $\underline{58.4}_{\pm0.3}$ | $64.3$ |
| Synthetic | Trait | Example+Wiki | $\underline{67.4}_{\pm0.3}$ | $51.2_{\pm1.0}$ | $\underline{68.3}_{\pm0.5}$ | $\underline{34.5}_{\pm0.9}$ | $53.0_{\pm3.1}$ | $\underline{81.2}_{\pm1.3}$ | $53.9_{\pm1.3}$ | $82.7_{\pm1.6}$ | $93.9_{\pm1.0}$ | $\mathbf{58.8}_{\pm0.4}$ | $\underline{64.5}$ |

generalization across other biological visual tasks arises from a larger training scale, as stated in the BIOCLIP 2 paper (Gu et al., 2025). Here, we use the same amount of data as BIOCLIP. Therefore, it is also understandable that no such "emergent properties" occur.

# F    DISCUSSION WITH RECENT WORK

This work proposes a new biological multimodal foundation model based on the combined supervision of taxonomic names and descriptive captions. This direction is complementary to recent advances that enrich biological models via scale or additional modalities. Gu et al. (2025) curate a large-scale TreeOfLife-200M dataset, with which they demonstrate that scaling hierarchical contrastive learning enables emergent properties. Gharaee et al. (2024) introduce DNA barcoding information as additional supervision. This work does not aim to replace scaling or other supervision sources, but focuses on an orthogonal direction of how synthetic captions help bridge biological images with multimodal foundation models. We demonstrate in this paper that descriptive captions, when grounded in biological knowledge, can significantly enhance the model's understanding of rich semantics and its species classification performance. The value of descriptive captions is, however, largely underexplored so far (Vendrow et al., 2024). It is also notable that the designed caption generation pipeline can be feasibly scaled up for larger datasets and integrated with more supervision dimensions through techniques like TaxaBind (Sastry et al., 2025).

A parallel line of work strengthens CLIP-style training by improving or expanding captions. FG-CLIP (Xie et al., 2025) constructs region-specific annotations, targeting fine-grained alignment capabilities. LaCLIP (Fan et al., 2023) uses LLMs to generate multiple versions of each caption according to format examples. VeCLIP (Lai et al., 2024) rewrites noisy web text into visual-enriched captions and mixes them with alt-text for training. CapsFusion (Yu et al., 2024) obtains descriptive captions from captioning models and refines them with large language models to achieve more semantically aligned supervision. These advances have pushed the frontier of general-domain multimodal foundation models and also enabled the pipeline of our approach. However, previous efforts mainly focus on general images, with limited exploration of scientific domains like organismal biology. When the pipeline is naively applied to biology, we empirically observe more hallucinations about biological details and oversight of discriminative traits. As discussed in §3, noisy captions can potentially harm the multimodal alignment. To this end, we explicitly inject the domain knowledge into the MLLM context. We demonstrate that the domain-specific contexts reduce hallucination and produce captions with more accurate biological details. Our work complements the gap for biological foundation models to explore the value of descriptive captions.

## G  BROADER APPLICABILITY TO SCIENTIFIC DOMAINS

A growing line of work aligns scientific images with domain-specific labels or short text surrogates (*e.g.*, object IDs, catalog entries, class names) to train multimodal representations (Parker et al., 2024; Vivanco Cepeda et al., 2023; Harnik et al., 2025). While effective for recognition and retrieval, this supervision is often *semantically thin*: many scientific labels are rare in everyday language, encode limited visual semantics by themselves, and are weak proxies for the causal or functional attributes that scientists care about. As a result, the learned alignment can overfit to dataset-specific correlates (instrument signatures, acquisition context, background cues) and provide limited support for higher-level tasks that require grounded interpretation.

BioCAP targets such a structural bottleneck in biodiversity. While images with taxonomic labels are abundant, these labels usually contain limited semantics and do not indicate traits of the species. *Instance-specific, attribute-rich text* is scarce and expensive to obtain. Similar gaps arise in many fields where visual observations are routinely collected, but natural-language reports are incomplete, non-standardized, or locked behind domain conventions. A naive approach is to directly generate captions at scale using general-purpose MLLMs; however, this can suffer from hallucination, which is shown in Figure 1, making synthetic supervision unreliable for scientific training.

The key idea of BioCAP is to convert *external scientific structure* into grounded language supervision. Concretely, we inject domain knowledge (Wikipedia-derived visual information) and curated format exemplars to constrain generation. This transforms open-ended captioning into a *constrained, schema-driven description* problem. The resulting captions are not intended to be free-form narratives; rather, they serve as reliable, attribute-oriented supervision that better matches how scientists record observations and how downstream models should reason.

This recipe generalizes beyond biodiversity to scientific domains that share three properties: (1) **large-scale imagery** (or image-like measurements), (2) **weak native text alignment** (IDs or labels that under-specify visual content), and (3) **available external structure** that can be encoded as constraints. Examples include: **medical imaging** (findings structured by radiology ontologies and reporting guidelines), **materials science and microscopy** (morphology, phases, defects, acquisition parameters), **chemistry and molecular imaging** (functional groups, conformations, assay conditions), **remote sensing** (land-cover attributes, phenology, sensor metadata, geospatial context), and **astronomy** (object properties, observation bands, instrument settings, survey metadata). In each case, schema-driven captioning can synthesize attribute-level text that is both grounded in the image and standardized across sources, enabling multimodal foundation models to learn representations that transfer to tasks requiring explanation, attribute retrieval, and hypothesis generation.

More broadly, BioCAP suggests a general pathway for scientific multimodal alignment: when paired image–text data is scarce or unreliable, *use domain constraints to manufacture trustworthy language supervision* can learn representations that are not only predictive, but also interpretable.

## H    HUMAN EVALUATION

### H.1    EVALUATION INSTRUCTION

---

**Evaluation Task and Criteria**

**Objective**
The goal of this study is to validate the quality and reliability of automatically generated captions. Your task is to independently evaluate them along predefined criteria. Specifically, we have sampled a set of examples, each paired with multiple candidate captions. For each example, you will review the captions and select the one that performs best under each evaluation criterion.

**Criterion-Level Assessment**
For each example, you will be provided with an image and three candidate captions. You are asked to evaluate the captions on four criteria:

- **Groundedness:** Does the caption align with features actually visible in the image?
- **Trait Specificity:** Does the caption highlight the most distinctive traits (e.g., color, patterns, morphology)?
- **Completeness:** Does the caption cover the 2-3 most salient visible aspects?
- **Clarity / Scientific Tone:** Is the caption written clearly, using precise and objective language?

For each criterion, select the caption that best satisfies the requirement.

**Notes**

- Each criterion is independent: you may choose different captions across criteria.
- Please read the criteria carefully and base your judgments only on the provided image and captions.
- Please only select the best caption per criterion.

---

**Participant Information and Consent**

**Thank you for agreeing to participate in this human evaluation study.** Before you begin, please carefully read the following information:

**Voluntary Participation**
Your participation in this study is entirely voluntary. You may stop at any time without penalty.

**Purpose**
The goal of this study is to evaluate the quality of automatically generated captions for biological images, using expert judgments across specific evaluation criteria.

**Data Collected**
Only your evaluation responses (e.g., which caption you select for each criterion) will be recorded.

**No Personal Information**
We will not collect any personally identifiable information (PII), such as your name, email address, or IP address. Your responses will remain anonymous.

**Confidentiality**
All data will be stored securely and used solely for research purposes. Results may be reported in aggregate form but will never be linked to individual evaluators.

**Risks and Benefits**
There are no anticipated risks associated with this study. While you may not receive direct personal benefit, your participation will contribute to the advancement of methods for evaluating scientific image descriptions.

---

> **Consent Statement**
> By clicking "Yes," you confirm that you have read the information above, agree to participate in this evaluation, and acknowledge that no personal data will be collected.

## H.2 EVALUATION STATISTICS

Table 21: Agreement between different human evaluators. Overall values are computed across all data (micro average).

| Attributes | Metric | |
| --- | --- | --- |
| | Raw Agreement | Gwet's AC1 |
| Groundedness | 66.5 | 0.509 |
| Specificity | 64.7 | 0.482 |
| Completeness | 78.8 | 0.724 |
| Clarity | 80.0 | 0.750 |
| Overall | 72.5 | 0.615 |

We conduct a human evaluation to assess the quality of captions generated by three different strategies: *Base*, *Trait*, and *Trait+Example+Wiki* (*Ours*). A total of 16 participants are involved in the study, including one ecologist and 15 computer science students. The evaluation covers 20 sets of images, each set containing 10 images, resulting in 200 images in total. For each image, we provide three candidate captions (one from each strategy) and ask evaluators to select the best one according to four evaluation attributes: *Groundedness*, *Specificity*, *Completeness*, and *Clarity*. Each image is independently evaluated by two different participants, ensuring two evaluation results.

We first report the win rate of each caption generation method, defined as the percentage of times a method's caption is selected as the best among the three candidates. As shown in Table 5, our full method (*Trait+Example+Wiki*) consistently outperforms both baselines across all four evaluation attributes, supporting the effectiveness of domain-specific contexts in improving the caption quality.

In addition, we assess the agreement between the two human evaluators. We report both the *raw agreement* (the proportion of samples where two evaluators select the same caption) and *Gwet's AC1* (Gwet, 2008), a chance-corrected agreement coefficient designed to provide stable estimates under imbalanced category distributions. Agreement is reported separately for each attribute, and we also report a micro-average across all data. As summarized in Table 21, both raw agreement (72.5% overall) and Gwet's AC1 (0.615 overall) indicate substantial inter-evaluator agreement, corresponding to the "substantial" level in standard interpretation scales (Landis & Koch, 1977).

## I DISCLOSURE OF LLM USAGE

Portions of this manuscript were polished for clarity and readability using an LLM. The LLM was not used to generate research ideas, design experiments, analyze data, or draw conclusions. All scientific content, methods, and results are the authors' original work.

