# OpenReview forum: "BioCAP: Exploiting Synthetic Captions Beyond Labels in Biological Foundation Models"
_ICLR.cc/2026/Conference — ICLR 2026 Poster_

### Official Review · Reviewer_qs4b · 2025-10-28

**Soundness:** 2
**Presentation:** 3
**Contribution:** 3
**Rating:** 4
**Confidence:** 5

**Summary:**

This paper proposes BioCAP, a biological multimodal model that leverages descriptive captions as an additional supervision signal alongside species labels. The authors argue that images and captions provide complementary views of a species’ latent morphospace, capturing biologically meaningful traits while suppressing spurious correlations. To overcome the scarcity of instance-level captions, they generate synthetic captions using a large multimodal language model (InternVL3 38B) guided by Wikipedia-derived visual information and taxon-tailored format examples. BioCAP is trained with a shared visual and text encoder, but uses dual visual projectors for taxonomic labels and captions. Experiments show BioCAP improves species classification and text–image retrieval with several baselines, while careful ablation studies demonstrate the benefits of their approach.

**Strengths:**

- **Novel approach**: The use of synthetic descriptive captions for biological multimodal models is innovative and addresses a clear bottleneck in organismal biology.
- **Strong empirical performance**: BioCAP outperforms multiple baselines (CLIP, SigLIP, BioTrove-CLIP, BioCLIP, FG-CLIP) on ten species classification benchmarks, retrieval tasks (PlantID, Cornell Bird), and natural language understanding (INQUIRE-Rerank).
- **Comprehensive evaluation**: The authors evaluate classification, retrieval, and language understanding, covering multiple organismal domains.
- **Human evaluation and careful ablation studies**: Caption quality is assessed using four metrics (groundedness, specificity, completeness, and clarity) showing the effectiveness of synthetic captions and addressing hallucination concerns. Ablations confirm that dual projectors outperform a single projector, and that adding synthetic captions improves generalization, even for species without Wikipedia coverage.

**Weaknesses:**

- **Choice of base model and baselines**: BioCAP is initialized from BIOCLIP, but the paper does not justify why **BIOCLIP 2**, the current state-of-the-art, was not used as the base model or included as a baseline, despite being cited in the experimentation section. Furthermore, there is no discussion of alternative multimodal models (e.g., LLaVA) or rationale for selecting BIOCLIP.
- **Evaluation scope and claims**: Evaluation is limited to zero-shot species classification and a subset of retrieval benchmarks; prior work includes few-shot tasks and additional biological visual datasets (FishNet, NeWT, AwA2, Herb. 19, PlantDoc).
- **Coverage limitations**: While ablation studies indicate improvements even for non-covered species, the paper does not explicitly analyze performance specifically on underrepresented species, leaving it unclear whether the approach benefits the full long-tail of species.
- **Trait diversity in curated examples**: Only up to three examples per taxonomic class may bias the model toward common traits, potentially underrepresenting rare or atypical characteristics. An ablation study on different number of curated examples would be a nice addition.
- **Propagation of synthetic caption errors**: While human evaluation supports caption quality, errors or omissions in synthetic captions could still affect model performance for species with limited or no reliable descriptions.

**Questions:**

**Questions for the Authors**

1. Why was BIOCLIP 2, the current state-of-the-art biological multimodal model, not used as the base model or included as a baseline?
2. Have the authors evaluated performance specifically on underrepresented or rare species to assess generalization across the long tail of biodiversity?
3. How sensitive is the model to the number and diversity of curated examples per taxonomic class? Would increasing this number improve generalization or trait coverage?
4. Are there analyses on how synthetic caption errors propagate through the training process, especially for species with limited textual coverage?

**Actionable Feedback**

1. Compare BioCAP against BIOCLIP 2, the current state-of-the-art, and provide justification for choosing BIOCLIP as the base model. Discussion of alternative multimodal foundation models (e.g., LLaVA) would also strengthen the paper.
2. Include few-shot classification, additional biological visual benchmarks (FishNet, Newt-AWA2, Herb-19, PlantDoc), and more diverse retrieval tasks. Explicit analysis of underrepresented or rare species would help assess generalization.
3. Assess whether limiting to three examples per taxonomic class biases the model, and consider ablations with more examples.

---

> ### Author Response · Authors · 2025-11-21
> **Rebuttal 1 of 2**
>
> Dear reviewer,
>
> We sincerely thank the reviewer for the constructive feedback and for recognizing our contributions in caption design and comprehensive empirical evaluation. The responses are listed below.
>
> > **1.  Choice of base model and baselines: BioCAP is initialized from BioCLIP, but the paper does not justify why BioCLIP 2, the current state-of-the-art, was not used as the base model or included as a baseline, despite being cited in the experimentation section. Furthermore, there is no discussion of alternative multimodal models (e.g., LLaVA) or rationale for selecting BioCLIP.**
> >
>
> Thank you for pointing this out. We clarify that BioCAP is initialized from the **same checkpoint of OpenAI CLIP** and trained with the **same images** as BioCLIP. This ensures a strictly fair comparison between training with class names only and with additional descriptive captions.
>
> We have an explicit discussion in Appendix F that our method demonstrates an orthogonal contribution to BioCLIP 2. BioCLIP 2 introduces more data (TreeOfLife-200M) into training, while we introduce additional supervision of descriptive captions into TreeOfLife-10M. The large training scale difference makes the direct comparison between BioCAP and BioCLIP 2 unfair. Note that BioCAP’s caption-based supervision could, in principle, be incorporated into BioCLIP 2 as well. We will treat this extension as an important future direction.
>
> We select to compare our model with BioCLIP instead of LLaVA based on the same reason that we intend to build a fair comparison between class names only and class names + descriptive captions. The same initialization (OpenAI CLIP) and the same training images ensure that the comparison between BioCAP and BioCLIP meets the requirement.
>
> > **2. Evaluation scope and claims: Evaluation is limited to zero-shot species classification and a subset of retrieval benchmarks; prior work includes few-shot tasks and additional biological visual datasets (FishNet, NeWT, AwA2, Herb. 19, PlantDoc).**
> >
>
> Thank you for raising the question. We report the results of few-shot classification in Table 18 and biological visual tasks in Table 19. We observe an overall improvement of BioCAP over BioCLIP in few-shot classification and a similar result in other biological visual tasks. We believe the smaller gap compared with zero-shot and multimodal retrieval is expected, given the nature of the added supervision. Compared with species names alone, descriptive captions introduce more "disturbances" that enrich the semantics carried by the embedding but also distract embeddings from the species prototypes. BioCAP yields a more interpretable intra-class structure tied to biological semantics. However, this supervision does not explicitly enforce more separation between visual embeddings of different species. Therefore, when captions provide better semantic organization and multimodal alignment, they do not contribute to better few-shot classification performances. On the other hand, the generalization across other biological visual tasks arises from a larger training scale, as stated in the BioCLIP 2 paper. Here, we use the same amount of data as BioCLIP. Therefore, it is also understandable that no such “emergent properties” occur. **We have added the results and discussions to Appendix E.5, Table 18 and Table 19, page 31.**
>
> > **3. Coverage limitations: While ablation studies indicate improvements even for non-covered species, the paper does not explicitly analyze performance specifically on underrepresented species, leaving it unclear whether the approach benefits the full long-tail of species.**
> >
>
> Thank you for this insightful suggestion. We have added additional experiments focusing explicitly on underrepresented species. We rank species by training sample count and define the bottom 5% as “few-image” species. Combining this with Wikipedia coverage, we obtain four categories:
>
> 1. Few-image + Non-covered
> 2. Few-image + Covered
> 3. Many-image + Non-covered
> 4. Many-image + Covered
>
> We split test data accordingly and report results for all four groups. Additionally, we evaluate on **species that never appeared in training** (the Rare Species benchmark). Across all splits, BioCAP consistently improves over BioCLIP. We have added the results to Appendix E.4, Table 17, page 30.
>
> | Model / Classification | Few-image + Non-covered | Few-image + Covered | Many-image + Non-covered | Many-image + Covered | Rare Species |
> | --- | --- | --- | --- | --- | --- |
> | CLIP | 16.2 | 32.3 | 21.1 | 23.4 | 25.7 |
> | BioCLIP | 54.0 | 30.4 | 35.4 | 42.1 | 37.6 |
> | BioCAP | **61.0** | **45.0** | **43.7** | **48.9** | **46.4** |

---

> ### Author Response · Authors · 2025-11-21
> **Rebuttal 2 of 2**
>
> > **4. Trait diversity in curated examples: Only up to three examples per taxonomic class may bias the model toward common traits, potentially underrepresenting rare or atypical characteristics. An ablation study on different number of curated examples would be a nice addition.**
> >
>
> Thank you for the question. We have extended the discussion on the format example design from the following three perspectives:
>
> **(1) Number of examples per class.**
>
> We vary the number of examples (1, 3, 5, 7). Using only one format example per taxonomic class limits the performance, while 3, 5, and 7 examples per class yield similar results. It shows that our curated set of up to three examples per class provides sufficient and stable information for caption generation.
>
> | Num of Examples | Classification | Image to Text | Text to Image | INQUIRE |
> | --- | --- | --- | --- | --- |
> | 1 | 33.2 | 52.7 | 53.2 | 33.9 |
> | 3 | 33.8 | **54.7** | 54.3 | **34.8** |
> | 5 | **34.0** | 54.3 | **54.7** | 34.5 |
> | 7 | 33.7 | 54.5 | 54.6 | 34.6 |
>
> **(2) Taxonomic level of examples.**
>
> We use taxonomic classes to prompt example generation in the paper. We also experiment with generating examples at the order level. The captions generated based on order-level and class-level examples show similar performances. Considering the storage and computational overhead for the format example generation, we use the class-level format examples for caption generation.
>
> | Taxonomic Level | Classification | Image to Text | Text to Image | INQUIRE |
> | --- | --- | --- | --- | --- |
> | Class | **33.8** | 54.7 | **54.3** | **34.8** |
> | Order | 33.2 | **55.1** | 53.9 | 33.6 |
>
> **(3) Stability across generation rounds.**
>
> We regenerate a second full set of format examples using Gemini and train BioCAP with them. Although the generated examples are different, the performance remains nearly identical, indicating that the example generation is robust across model runs.
>
> | Generation Round | Classification | Image to Text | Text to Image | INQUIRE |
> | --- | --- | --- | --- | --- |
> | 1 | **33.8** | 54.7 | 54.3 | **34.8** |
> | 2 | 33.6 | **54.9** | **54.5** | 34.7 |
>
> **We have included these results and a more detailed description of the example pipeline in Appendix E.3, page 30.**
>
> > **5. Propagation of synthetic caption errors: While human evaluation supports caption quality, errors or omissions in synthetic captions could still affect model performance for species with limited or no reliable descriptions.**
> >
>
> We acknowledge this concern. As detailed in Appendix Table 6, at the family level, 96.1% of samples have at least one species covered by Wikipedia within the same family. Therefore, while some species are not directly supported by Wikipedia-derived visual descriptions, they still benefit from generalized knowledge of similar classes during training. While we cannot guarantee that our pipeline is free of caption noise, we observe qualitatively that the generated captions contain much less hallucination. The quantitative results also support that the generated captions improve the multimodal alignment between images and their class names, including those species with no Wikipedia-derived visual descriptions.

---

> > ### Comment · Reviewer_qs4b · 2025-11-24
> >
> > I thank the authors for their response and appreciate the additional analyses on coverage and trait diversity. I also agree that using BioCLIP rather than BioCLIP2 is fair, and I accept the assumption that “a smaller gap compared with zero-shot and multimodal retrieval is expected, given the nature of the added supervision.” However, the results still show that in nearly half of the cases, BioCLIP outperforms BioCAP when both are trained with only a few samples from the target dataset. This suggests a potential limitation in BioCAP’s generalization. Yet, since maintaining fairness with BioCLIP means that this issue cannot be addressed simply by adding more samples, it may instead require improving the quality of the synthetic captions.
> >
> > Given that this concern remains unresolved, I cannot raise my rating above a 6.

---

> > > ### Author Response · Authors · 2025-11-26
> > > **Response to the new questions**
> > >
> > > We thank the reviewer for the thoughtful engagement and for updating the score. We also appreciate the constructive hypothesis regarding the relationship between caption quality and generalization.
> > >
> > > Regarding the few-shot performance, we would like to clarify the evaluation protocol and share a key finding from our new experiments. First, our few-shot evaluation is entirely training-free. We utilize a standard nearest-neighbor approach based on class prototypes within the frozen embedding space, without any parameter updates. Therefore, the few-shot performance is purely dependent on the embedding distribution.
> > >
> > > As shown in the Table below, we compared models trained with different caption generation strategies. The results demonstrate that while higher-quality captions significantly boost zero-shot and retrieval metrics (shown in Table 5), the few-shot performance shows minimal variance across different strategies. This suggests that few-shot capability is largely decoupled from caption quality. We sincerely appreciate the reviewer’s insight. While our results indicate that caption quality is not the primary bottleneck for the current few-shot performance, we agree that refining caption quality and exploring methods to further enhance few-shot generalization are exciting and critical directions for future research. **We have added the results to Appendix E.5 Table 20, page 32.**
> > >
> > >
> > >
> > > **Table: One-Shot and Five-Shot classification accuracy (mean ± std) across varying caption generation strategies and datasets.**
> > >
> > > One-Shot Classification
> > >
> > > | **Caption** | **Prompt** | **Context** | **NABirds** | **Plankton** | **Insects** | **Insects 2** | **Camera Trap** | **PlantNet** | **Fungi** | **PlantVillage** | **Med. Leaf** | **Rare Species** | **Mean** |
> > > | --- | --- | --- | --- | --- | --- | --- | --- | --- | --- | --- | --- | --- | --- |
> > > | None | - | - | **45.1**±0.9 | 29.9±1.2 | **45.6**±1.5 | 18.7±0.6 | 33.3±3.6 | **64.1**±3.7 | **35.8**±1.8 | 59.8±2.7 | 78.0±1.7 | 36.9±0.5 | 44.7 |
> > > | Wiki Page | - | - | 40.8±1.0 | 31.9±1.7 | 42.4±1.6 | 19.5±1.0 | **35.3**±2.5 | 60.7±3.9 | 35.0±0.6 | 58.9±3.4 | **82.1**±1.7 | **38.1**±0.7 | 44.5 |
> > > | Synthetic | Base |  | 38.6±0.8 | **34.3**±0.6 | 40.3±1.5 | **20.0**±0.7 | 34.6±3.4 | 58.1±2.2 | 34.0±2.1 | **62.2**±3.9 | 81.9±2.7 | 35.9±0.5 | 44.0 |
> > > | Synthetic | Trait |  | 41.5±0.8 | 29.9±1.4 | 42.1±1.2 | 19.5±0.9 | 34.3±3.2 | 63.1±3.6 | 35.3±1.8 | 60.9±3.3 | 81.1±2.1 | 37.2±0.8 | 44.5 |
> > > | Synthetic | Trait | Example | 41.9±0.7 | 31.5±1.3 | 43.5±1.4 | 19.3±0.8 | 34.5±2.9 | **64.1**±4.3 | 34.9±1.3 | 61.5±3.1 | 79.6±2.1 | 36.8±0.4 | 44.8 |
> > > | Synthetic | Trait | Example+Wiki | **43.3**±0.9 | 31.8±1.6 | 43.5±0.7 | 19.3±1.0 | 33.8±4.6 | 63.5±0.6 | 35.6±2.9 | 61.6±1.6 | 78.7±0.7 | 37.8±0.7 | **44.9** |
> > >
> > >
> > > Five-Shot Classification
> > >
> > > | **Caption** | **Prompt** | **Context** | **NABirds** | **Plankton** | **Insects** | **Insects 2** | **Camera Trap** | **PlantNet** | **Fungi** | **PlantVillage** | **Med. Leaf** | **Rare Species** | **Mean** |
> > > | --- | --- | --- | --- | --- | --- | --- | --- | --- | --- | --- | --- | --- | --- |
> > > | None | - | - | 67.0±0.4 | **55.7**±0.6 | 65.0±0.4 | **34.8**±0.6 | 52.4±2.2 | 78.3±1.0 | 50.6±1.1 | **84.2**±0.9 | **95.8**±0.8 | 57.1±0.3 | 64.1 |
> > > | Wiki Page | - | - | **69.0**±0.3 | 49.2±0.7 | **69.9**±0.7 | 33.4±0.6 | 53.5±1.2 | **82.0**±1.1 | 54.6±1.0 | 79.6±0.7 | 93.3±0.3 | 57.9±0.3 | 64.3 |
> > > | Synthetic | Base |  | 66.2±0.3 | 55.0±0.8 | 67.3±0.7 | 34.5±0.4 | **55.3**±2.0 | 78.9±1.4 | **54.8**±1.2 | 82.5±1.0 | 93.6±1.0 | 58.2±0.3 | **64.6** |
> > > | Synthetic | Trait |  | 67.0±0.3 | 49.7±0.5 | 67.7±0.3 | 34.5±0.7 | 52.7±2.4 | 80.2±0.6 | 53.9±1.1 | 83.2±0.8 | 94.5±0.6 | 58.2±0.4 | 64.2 |
> > > | Synthetic | Trait | Example | 67.0±0.3 | 51.6±0.4 | 68.1±0.6 | 33.9±0.3 | 53.3±2.1 | 81.0±1.0 | 54.5±1.1 | 81.8±1.2 | 93.2±0.8 | 58.4±0.3 | 64.3 |
> > > | Synthetic | Trait | Example+Wiki | 67.4±0.3 | 51.2±1.0 | 68.3±0.5 | 34.5±0.9 | 53.0±3.1 | 81.2±1.3 | 53.9±1.3 | 82.7±1.6 | 93.9±1.0 | **58.8**±0.4 | 64.5 |

---

### Official Review · Reviewer_uX5T · 2025-10-31

**Soundness:** 3
**Presentation:** 3
**Contribution:** 3
**Rating:** 8
**Confidence:** 4

**Summary:**

The authors create synthetic instance-level captions of images from the TreeOfLife-10M dataset and use them to train a version of BioCLIP with a caption encoder. The captions are morphological trait-based descriptions of the animals that correspond to subregions in the images. The authors articulate their synthetic caption generation strategy, report benchmarks on classification and retrieval, and perform a detailed ablation study. The results indicate that the trait captions improve classification and help constrain model attention to animal parts that help with classification.

**Strengths:**

- Originality: the authors use an interesting approach to caption generation, prompting MLLMs with format specification on a per-class basis. The idea was to constrain output to focus on salient morphological description that can be challenging to pick out of raw text without guidance. The approach to leveraging the captions in BioCLIP is appealingly simple.
- Quality: The overall qualtiy is quite good, lots of experiments executed with sufficient data. The ablation studies do a nice job of illustrating the benefits of adding these captions to the model.
- Clarity: The paper is well-written and relatively easy to follow. There are a few sections (noted below) that could benefit from some rewriting.
- Significance: This work provides more evidence that species classification is easier when you can direct the model to pay more attention to informative parts of the target. The use of captions, and the way in which they were generated, is a useful addition to the toolkit.

**Weaknesses:**

The format example design discussion needs expansion. This seems to be a critical element of the work and it is treated very narrowly. It isn't clear what the 'classes' are that were used to query Gemini Deep Research, who did the winnowing of the results, or how consistent the examples were across the classes. This element may itself benefit from an exploration of how variable those exemplars are between model runs and how consistent the human overseers where in selecting appropriate descriptions.

**Questions:**

- What is the taxonomic breakdown of the resulting descriptions? At line 224 you indicate the descriptions cover ~32% of the species in TreeOfLife-10M. Are there any biases in how the generated captions or is it uniformly distributed across organism groups?
- What are the limitations of the caption generation strategy? Is the 32% coverage a fundamental limit of what is available on Wikipedia?
- Are there any potential issues when mapping genus level descriptions onto species? Does that result in multiple species having the same descriptions?
- Can you elaborate on the human evaluation task? Per the supplement, there was only one domain expert in the group of annotators and 15 computer science students. How well prepared are your human validators to assess the quality of the captions? What is the ecologist's area of expertise?
- What is the overlap in the species coverage in the zero-shot datasets? How much of the performance improvement is related to prior knowledge of the organism groups being tested?
- At line 230, it is noted that there are 347 taxonomic classes in TreeOfLife-10M. Is that a typo? Or are classes in this case the level of the taxonomic tree?

---

> ### Author Response · Authors · 2025-11-21
> **Rebuttal part 1 of 2**
>
> Dear reviewer,
>
> We sincerely thank the reviewer for the constructive feedback and for recognizing our contributions in caption design and comprehensive empirical evaluation. The responses are listed below.
>
> > **W. The discussion of format example design is insufficient. It is unclear what the “classes” are, how exemplars were selected or filtered, and how consistent they were across classes or model runs. More analysis is needed to show the stability and variability of these examples.**
> >
>
> Thank you for highlighting this concern. The “classes” in format examples refer to biological taxonomic classes (e.g., Aves, Insecta, Mammalia).
>
> We have extended the discussion on the format example design from the following three perspectives:
>
> **(1) Number of examples per class.**
>
> We vary the number of examples (1, 3, 5, 7). Using only one format example per taxonomic class limits the performance, while 3, 5, and 7 examples per class yield similar results. It shows that our curated set of up to three examples per class provides sufficient and stable information for caption generation.
>
> | Num of Examples | Classification | Image to Text | Text to Image | INQUIRE |
> | --- | --- | --- | --- | --- |
> | 1 | 33.2 | 52.7 | 53.2 | 33.9 |
> | 3 | 33.8 | **54.7** | 54.3 | **34.8** |
> | 5 | **34.0** | 54.3 | **54.7** | 34.5 |
> | 7 | 33.7 | 54.5 | 54.6 | 34.6 |
>
> **(2) Taxonomic level of examples.**
>
> We use taxonomic classes to prompt example generation in the paper. We also experiment with generating examples at the order level. The captions generated based on order-level and class-level examples show similar performances. Considering the storage and computational overhead for the format example generation, we use the class-level format examples for caption generation.
>
> | Taxonomic Level | Classification | Image to Text | Text to Image | INQUIRE |
> | --- | --- | --- | --- | --- |
> | Class | **33.8** | 54.7 | **54.3** | **34.8** |
> | Order | 33.2 | **55.1** | 53.9 | 33.6 |
>
> **(3) Stability across generation rounds.**
>
> Note that we intend to capture different properties across classes in the generated format examples, as the model should focus on different traits when describing different species. The structured prompting strategy itself ensures the style consistency between different format examples. Regarding the stability of the generated format examples across different runs, we regenerate a second full set of format examples using Gemini and train BioCAP with them. Although the generated examples are different, the performance remains nearly identical, indicating that the example generation is robust across model runs.
>
> | Generation Round | Classification | Image to Text | Text to Image | INQUIRE |
> | --- | --- | --- | --- | --- |
> | 1 | **33.8** | 54.7 | 54.3 | **34.8** |
> | 2 | 33.6 | **54.9** | **54.5** | 34.7 |
>
> After obtaining candidate descriptions from Gemini Deep Research, two evaluators with biology and CS interdisciplinary backgrounds filtered the outputs. They removed unreliable or off-topic content and retained only descriptions of the intended morphological focus.
>
> **We have included these results and a more detailed description of the example pipeline in Appendix E.3, page 30.**

---

> > ### Author Response · Authors · 2025-11-21
> > **Rebuttal part 2 of 2**
> >
> > > **Q1**.  **What is the taxonomic breakdown of the resulting descriptions? At line 224 you indicate the descriptions cover ~32% of the species in TreeOfLife-10M. Are there any biases in how the generated captions or is it uniformly distributed across organism groups?**
> > >
> >
> > Thank you for the question. As detailed in Appendix Table 6, Wikipedia-derived descriptions are naturally uneven across organism groups. However, at the family level, 96.1% of samples have at least one species covered by Wikipedia within the same family. Therefore, while some species are not directly supported by Wikipedia-derived visual descriptions, they still benefit from generalized knowledge of similar classes during training.
> >
> > > **Q2. What are the limitations of the caption generation strategy? Is the 32% coverage a fundamental limit of what is available on Wikipedia?**
> > >
> >
> > Yes, the primary limitation is the incompleteness of Wikipedia: many species lack detailed morphological descriptions. In principle, additional knowledge sources (e.g., field guides, ecological databases, taxonomic literature) could be incorporated to increase coverage of domain-specific knowledge. We will emphasize this as an important direction for future work.
> >
> > > **Q3. Are there any potential issues when mapping genus level descriptions onto species? Does that result in multiple species having the same descriptions?**
> > >
> >
> > Thank you for raising this point. This mapping design is biologically reasonable: in taxonomy, closely related species within a genus typically share most of their diagnostic morphological characters, differing only in a small subset of traits [1].
> >
> > We acknowledge that this fallback strategy can result in multiple species receiving the same Wikipedia-derived visual descriptions. However, the visual descriptions are not used directly as captions, but rather as context to guide caption generation. The MLLM still relies on the input image to produce a species-specific synthetic caption. Our qualitative analysis in Figure 8 shows that, even when the genus-level prompt is shared, the generated captions differ.
> >
> > **We have added clarifying text and supporting qualitative examples in the revision,Appendix, B.4, Figure 8, page 22.**
> >
> > > **Q4. Can you elaborate on the human evaluation task? Per the supplement, there was only one domain expert in the group of annotators and 15 computer science students. How well prepared are your human validators to assess the quality of the captions? What is the ecologist's area of expertise?**
> > >
> >
> > Our human evaluation involves evaluators instead of ground-truth taxonomic labelers. The evaluation criteria (groundedness, specificity, completeness, clarity) focus on whether the textual description aligns with the visible content of the image. This assessment does not require deep biodiversity expertise, as it primarily involves verifying visual-textual correspondence. We have one biologist evaluator, whose areas of expertise are beetles and forest diversity. The remaining evaluators have combined backgrounds in biology and computer science, making them well-suited for evaluating visual-textual alignment.
> >
> > > **Q5**. **What is the overlap in the species coverage in the zero-shot datasets? How much of the performance improvement is related to prior knowledge of the organism groups being tested?**
> > >
> >
> > Thank you for this question. Rare Species is deliberately excluded from TreeOfLife-10M, as stated in the BioCLIP paper. We haven’t explicitly checked the other benchmarks, as they might have different naming rules from the training set. We suppose all other species in our zero-shot evaluation sets are present in the training dataset.
> >
> > Our ablations on Wikipedia-covered vs. non-covered species in Table 4 show that BioCAP improves generalization even when species are not supported by Wikipedia-derived visual descriptions. The improvements for these non-covered species are not due to prior exposure to domain-specific knowledge, but rather the generalization across species.
> >
> > > **Q6. At line 230, it is noted that there are 347 taxonomic classes in TreeOfLife-10M. Is that a typo? Or are classes in this case the level of the taxonomic tree?**
> > >
> >
> > The “taxonomic class” refers to the biological taxonomic rank (e.g., Mammalia, Insecta, Aves).
> >
> > [1] Mayr, E. Principles of Systematic Zoology. McGraw-Hill, 1991.

---

### Official Review · Reviewer_KyEW · 2025-11-01

**Soundness:** 3
**Presentation:** 4
**Contribution:** 3
**Rating:** 6
**Confidence:** 5

**Summary:**

This work present a biological multimodal foundation model that integrates synthetic, descriptive captions as additional supervision beyond taxonomic labels to improve species understanding. The authors propose to generate synthetic instance-level captions using multimodal LLMs guided by Wikipedia-derived visual information and taxon-specific format exam, then train a model with dual text views(species name + caption, )and dual visual projectors to decouple supervision. The results shown more than 20% improvements across 10 classification tasks.

**Strengths:**

- Multimodal alignment in biology is an under-explored and important task.
- The paper propose an interesting idea that images and captions are treated as complementary projections of a species’ latent morphospace, so aligning them helps capture diagnostic traits while suppressing noise.
- The paper introduced a dual-projector architecture elegantly separates taxonomy vs. caption supervision, and use context-guided caption generation (Wikipedia + format examples) effectively mitigates LLM hallucination.
- The evaluation is comprehensive, including multiple benchmarks, both quantitative and qualitative analysis.

**Weaknesses:**

- The caption reliance on Wikipedia-derived descriptors could reinforce taxonomic bias and exclude rare or poorly documented species. How does the framework handle species without any Wikipedia entry or minimal trait descriptions?
- The repeated LLM re-generation could produce inconsistent style or attribute focus across species, causing the semantic drift.
- Potential scalable issue since the caption generation and derail-view training are computational costly.

**Questions:**

- Is there any caption noise? how did you filter those?
- What effort is needed to extend the framework to video data?
- The method is specifically designed for biology domain. How is the generalization capability to other scientific domains?
- Is there any quantitative result for Grad-CAM?
- Who are the human annotator? are they biology expert?

---

> ### Author Response · Authors · 2025-11-21
> **Rebuttal part 1 of 2**
>
> Dear reviewer,
>
> We sincerely thank the reviewer for the constructive feedback and for recognizing our contributions in caption design and comprehensive empirical evaluation. The responses are provided below.
>
> > **W1. The caption reliance on Wikipedia-derived descriptors could reinforce taxonomic bias and exclude rare or poorly documented species. How does the framework handle species without any Wikipedia entry or minimal trait descriptions?**
> >
>
> Thank you for raising this concern. Rare or poorly documented species are still handled through LLM-based caption generation, even without Wikipedia entries. Table 4 presents an ablation comparing covered vs. non-covered species, showing that BioCAP improves generalization even for species without Wikipedia descriptions. This suggests that incorporating captions benefits the long tail rather than reinforcing only the Wikipedia-covered species.
>
> > **W2. The repeated LLM re-generation could produce inconsistent style or attribute focus across species, causing the semantic drift.**
> >
>
> We appreciate this point. In BioCAP, we do not perform repeated regeneration. Each species receives **one caption**.
>
> For the attribute focus, we design format examples tailored for different taxonomic classes, as they naturally have different diagnostic attributes. Therefore, differences in emphasized traits across species are intentional.
>
> > **W3. Potential scalable issue since the caption generation and derail-view training are computational costly.**
> >
>
> Thank you for raising this point. We would like to clarify that the computational cost of our pipeline is comparable to a standard model training run, rather than significantly higher, as shown in Appendix B.2. The caption generation process can be scaled efficiently through distributed inference.
>
> Dual-projector training adds negligible overhead compared with the original single-projector design, as only one lightweight linear layer is added.
>
> Overall, the framework remains practically scalable, and both caption generation and training can accommodate much larger datasets simply by increasing the GPU count or extending the runtime.

---

> ### Author Response · Authors · 2025-11-21
> **Rebuttal part 2 of 2**
>
> > **Q1. Is there any caption noise? how did you filter those?**
> >
>
> While we cannot guarantee that all the caption noise is eliminated, our pipeline substantially reduces the noise through:
>
> - Visual grounding from Wikipedia-derived descriptions
> - Structured format examples tailored for each taxonomic class
> - Trait-focused prompting
>
> These elements collectively suppress hallucination and promote attribute consistency. We include representative examples in Figure 4 to demonstrate much more grounded captions. The results in Table 3 also suggest a much better quality of the captions generated by our pipeline.
>
> > **Q2. What effort is needed to extend the framework to video data?**
> >
>
> Thank you for the thoughtful question. While our work currently focuses on static images, the framework extends naturally to video. The guided caption generation process (using format examples and Wikipedia-derived descriptions) can be applied directly with an MLLM that supports video understanding, and the resulting captions can be aligned with video embeddings under the same dual-view objective. In practice, the only change needed is replacing the image encoder with a video encoder, making the adaptation straightforward. We view this as an exciting direction for future work.
>
> > **Q3. The method is specifically designed for biology domain. How is the generalization capability to other scientific domains?**
> >
>
> This is an insightful question. Many scientific domains provide only **class names** without rich annotations. Our approach demonstrates that descriptive captions, when grounded in domain knowledge, provide essential supervision that aligns domain-specific images with their labels. This insight is domain-agnostic. For example:
>
> - Materials science (crystal phase descriptions)
> - Medicine (pathology findings)
> - Geology (rock morphology)
>
> The core idea is broadly applicable: using expert-informed textual supervision to enrich multimodal alignment.
>
> > **Q4. Is there any quantitative result for Grad-CAM?**
> >
>
> Yes. We conducted a quantitative evaluation on the CUB [1] dataset, which provides bounding boxes for each image. Using species names as text prompts, we compute Grad-CAM for CLIP, BioCLIP, and BioCAP across the entire test set, then evaluate them using the energy-based pointing game [2]. The energy-based pointing game measures the proportion of Grad-CAM activation that falls inside the ground-truth bounding box.
>
> BioCAP achieves higher localization scores than both CLIP and BioCLIP, as shown in the table below, demonstrating better alignment between model attention and fine-grained biological concepts. This improvement is aligned with the improved classification and retrieval performance listed in the paper. **We have added the results to Appendix Table 14, page 29.**
>
> | Model | Localization |
> | --- | --- |
> | CLIP | 0.36 |
> | BioCLIP | 0.43 |
> | BioCAP | **0.47** |
>
> > **Q5.  Who are the human annotator, are they biology expert?**
> >
>
> We have human evaluators instead of ground-truth annotators for the caption-quality assessment, among whom there is one biologist.
>
> [1] Wah, C et al. The Caltech-UCSD Birds-200-2011 Dataset. California Institute of Technology. 2011.
>
> [2] Wang, H et al. Score-CAM: Score-weighted Visual Explanations for Convolutional Neural Networks. CVPR Workshops. 2020.

---

### Official Review · Reviewer_XfBu · 2025-11-01

**Soundness:** 2
**Presentation:** 3
**Contribution:** 3
**Rating:** 6
**Confidence:** 2

**Summary:**

The paper proposes a BIOCLIP-style model that, during training, aligns images with taxonomic labels and instance-specific synthetic captions generated by an MLLM. The captions are constrained by Wikipedia-derived visual descriptions and taxon-tailored format examples to reduce hallucinations. The evaluation show strong performance for species classification and text-image retrieval by training on TreeOfLife-10: +8.8% average top-1 over BIOCLIP across 10 zero-shot species classification sets, and +21.3% average on natural-language tasks/retrieval. Ablations show that human eval favors the proposed model's captions over others.
The qualitative analysis with Grad-CAM images and t-SNE embeddings further highlight the improved embeddings with BioCAP.

**Strengths:**

The paper proposes interesting tweaks like using images and captions as complementary views of a species’ latent trait vector and training with contrastive learning to emphasize diagnostic features. The dual projector cleanly separates heterogeneous supervision as compared to a single projector.

The paper will be of reasonable interest for the broader community interested in training VLMs for life sciences.

**Weaknesses:**

Captions are biased toward the chosen InternVL3-38B; and there is no cross-MLLM comparison.
Behavior labels for analysis are auto-assigned by GPT-4o that may cause label drift.
Large deltas are shown in experimental results, but statistical significance/confidence intervals aren’t reported for benchmarks.

**Questions:**

can caption-aligned features also help zero-shot retrieval/classification?
How do results change with different captioners?
What is the dedup protocol used? How do you ensure held-out test set?
Are there any cases of observed hallucination in the final captions?

---

> ### Author Response · Authors · 2025-11-21
> **Rebuttal part 1 of 2**
>
> Dear reviewer,
>
> We sincerely thank the reviewer for the constructive feedback and for recognizing our contributions in caption design and comprehensive empirical evaluation. Please find the responses below.
>
> > **W1. Captions are biased toward the chosen InternVL3-38B. And there is no cross-MLLM comparison.**
> >
>
> Thank you for raising this important point. To evaluate whether BioCAP relies heavily on a particular captioner, we conducted additional experiments using three different MLLMs: LLaVA-next-34B, Qwen2.5-VL-32B, and InternVL3-38B. And three model sizes for internvl3 (7B, 38B, 78B), results show in the following table. We summarize our observations below:
>
> - BioCAP trained using captions from Qwen2.5-VL-32B and InternVL3-38B achieves nearly identical performance, indicating that the method is robust to the choice of captioner.
> - LLaVA-generated captions lead to slightly worse classification performances, which might be due to their comparatively weaker visual grounding and instruction-following capabilities.
> - For the model size, the 8B model performs modestly worse, while 38B and 78B produce almost indistinguishable results.
>
> These findings indicate that our caption-generation pipeline does not depend on highly specialized or extremely large MLLMs.
>
>  **We have included a cross-MLLM comparison in the revised version, Table 7, page 8.**
>
> | Generators | Classification | Image to Text | Text to Image | INQUIRE |
> | --- | --- | --- | --- | --- |
> | LLaVA-NexT-34B | 32.9 | 54.1 | 54.7 | 34.4 |
> | Qwen2.5-VL-32B | **34.1** | 53.3 | **55.5** | **35.2** |
> | InternVL3-38B | 33.8 | **54.7** | 54.3 | 34.8 |
>
> | Model Size | Classification | Image to Text | Text to Image | INQUIRE |
> | --- | --- | --- | --- | --- |
> | InternVL3-8B | 31.7 | 49.4 | 50.2 | 33.7 |
> | InternVL3-38B | 33.8 | 54.7 | 54.3 | **34.8** |
> | InternVL3-78B | **33.9** | **54.8** | **55.6** | 34.5 |
>
> > **W2. Behavior labels for analysis are auto-assigned by GPT-4o that may cause label drift.**
> >
>
> Thank you for the question. In addition to the annotations of GPT-4o, we also collected labels from Gemini 2.5 pro and humans. We show the agreement ratio in the following table:
>
> | Comparison | Agreement (%) |
> | --- | --- |
> | GPT-4o vs Human | 95.1 |
> | Gemini vs Human | 92.9 |
> | GPT-4o vs Gemini | 94.0 |
>
> These results show that behavior annotation is a robust and low-ambiguity task, with two independent MLLMs producing labels consistent with human judgment.
>
> **We have included this validation study in Appendix, Table 11, page 23.**
>
> > **W3. Large deltas are shown in experimental results, but statistical significance/confidence intervals aren’t reported for benchmarks.**
> >
>
> Thank you for raising this point. Our core evaluations, including zero-shot classification and zero-shot retrieval, are fully deterministic because they use a fixed model for feature extraction and a fixed similarity-computation pipeline. For this reason, statistical confidence intervals are not directly applicable to the benchmarks. We use another way of re-training BioCAP with a different group of format examples. The results are presented in Appendix Table 16. While trained with completely different captions, the resulting models demonstrate similar performances, indicating the statistical significance of our entire pipeline.

---

> ### Author Response · Authors · 2025-11-21
> **Rebuttal part 2 of 2**
>
> > **Q1. Can caption-aligned features also help zero-shot retrieval/classification? How do results change with different captioners?**
> >
>
> Yes. This is precisely the central finding of BioCAP. We first analyze in Section 3.1 that “captions that faithfully reflect visible and potentially diagnostic characters of the species” help the model build more accurate species prototypes. Empirically, we observe:
>
> - In Table 1 (zero-shot species classification), BioCAP consistently outperforms all baselines, including CLIP, SigLIP, FG-CLIP, BioTrove-CLIP, and BioCLIP.
> - In Table 2 (retrieval), BioCAP also yields substantial gains on PlantID, Cornell Bird, and INQUIRE.
>
> Regarding captioners, we show results in W1 that captioners from different model families (LLaVA, Qwen2.5-VL, InternVL3) all produce significant performance improvements.
>
> > **Q2. What is the dedup protocol used? How do you ensure held-out test set?**
> >
>
> Thank you for this question. We use TreeOfLife-10M for training, and we directly adopt the classification benchmarks from the BioCLIP/BioCLIP 2 paper (NABirds, Meta-Album, Rare Species, Camera Trap). These test sets are held out during the construction of the TreeOfLife-10M dataset. Please refer to their original papers for more details. For the other datasets we used:
>
> - INQUIRE is constructed to avoid overlap with iNat21, the only iNaturalist subset in TreeOfLife-10M.
> - The images of the other two retrieval datasets are mainly uploaded individually instead of being systematically collected. There is a small possibility that they overlap with EOL. Therefore, we further run a duplication check for these two datasets. The results suggest that only 4.1% and less than 0.1% of them are in the training set for PlantID and Cornell Bird, respectively. We have excluded the overlapping images and re-evaluated all the models on retrieval benchmarks. The new results are consistent with our previous conclusions.
>
> **We have added a more explicit clarification of these deduplication processes in the revision, Appendix, D.3, page 25** .
>
> > **Q3. Are there any cases of observed hallucination in the final captions?**
> >
>
> While we cannot guarantee that the final captions do not contain any hallucination, our pipeline significantly reduces hallucination via:
>
> - Wikipedia-derived visual descriptions
> - Format examples tailored to taxonomic classes
> - Strict prompting focused on morphological traits
>
> As shown in Table 5, the human evaluation finds the captions to have much better groundedness, specificity, completeness, and clarity. By manually examining a few generated captions, we do not notice obvious hallucinations.

---

### Author Response · Authors · 2025-12-03
**General Response**

We sincerely thank all reviewers (R1: XfBu, R2: KyEW, R3: uX5T, R4: qs4b) for their constructive feedback. During the rebuttal period, we have addressed concerns and added extensive additional experiments. We summarize the key updates below:

**1. Does BioCAP depend on the specific architecture or size of the caption generator? (R1 W1)** We verified that the best performances can be achieved with different MLLM backbones (Qwen2.5, InternVL3) and model sizes (38B to 78B). This confirms that our pipeline does not depend on highly specialized or extremely large MLLMs. **(new results in Table 7, page 8)**

**2. Is the format example design robust to variations in quantity, taxonomic level, and generation runs? (R3 W1, R4 W)** We conducted extensive ablations to demonstrate that downstream model performance remains stable across varying numbers of examples (3, 5, or 7), different taxonomic levels (Class vs. Order), and independent caption generation runs. This confirms the reliability of the caption generation pipeline and the consistency of the resulting supervision. **(new results in Table 6, page 8; Tables 15 & 16)**

**3. Does the synthetic caption pipeline suffer from significant hallucination or noise? (R1 Q3, R2 Q1)** We emphasized that our pipeline, leveraging Wikipedia-derived descriptions and format examples, significantly suppresses hallucination. Human evaluation (Table 5, page 7) and qualitative inspection confirm that the generated captions are well-grounded, minimizing the risk of noise affecting model training.

**4. Does the limited Wikipedia species coverage (~32%) restrict knowledge transfer or bias the model? (R3 Q1/Q2, R4 W)** We clarified that **96.1%** of training samples benefit from family-level coverage, enabling effective knowledge transfer even when species-level descriptions are missing. Incorporating additional knowledge sources is identified as a key future direction to further improve coverage.

**5. Does BioCAP generalize to long-tail and underrepresent species? (R3 Q5, R4 W)** We split the test species into four subsets based on the number of training images and Wikipedia coverage (e.g., "few-image + non-covered"). BioCAP consistently outperforms BioCLIP across **all groups**, confirming robust improvements for the **entire long tail** regardless of data scarcity. **(new results in Table 17, page 30)**

We explicitly thank R4 for the thoughtful engagement throughout the process. The reviewer’s most recent response acknowledged that we had addressed most of the concerns and raised the score. The only remaining concern was the lower performance improvement in the few-shot setting. The reviewer suggested that improving caption quality might address the few-shot limitation. We conducted additional ablations comparing the few-shot performance across different caption strategies. The results demonstrate that while lower-quality captions significantly degrade zero-shot and retrieval metrics (Table 4, page7), their few-shot performance remains stable. This finding indicates that few-shot capability is **largely decoupled from caption quality** **in this setting**. And we agree that refining caption quality and exploring methods to further enhance few-shot generalization are exciting and critical directions for future research. **(new results in Table 18, 19 & 20, page 30-31)**

---

### Meta-Review · Area_Chair_ZUPc · 2026-01-10

**Summary:**

The paper provides a method for contextualizing synthetic data generation for specific domains (biological in this case), and therefore questions about the details of synthetic generation/pipeline, generality of claims/results across model types, and transferability of the learned models to other (non-zero-shot) tasks were all legitimate concerns raised by the reviewers. Specifically, Reviewer XfBu raised concerns about potential caption biases towards the model utilized and potential GPT-4o auto-labeling bias which is a valid concern. Similarly, Reviewer KyEW mentioned other potential biases or specificity towards the chosen domain (e.g. taxonomic bias and applicability to other scientific domains). Reviewer uX5T raised concerns regarding lack of details in terms of the format example design. Finally, Reviewer qs4b mentions use of the models for other downstream tasks such as few-shot classification, which is typical in the literature.

In all, the rebuttal addresses many of these concerns (and others) in detail, presenting a number of results, ablations, and textual responses.

**Reviewer Concerns:**

Most of the above reviewer concerns were addressed. Some outstanding concerns include additional details about the format example design (e.g. who did the winnowing of the results, etc.) which I could not find in E.3 which is quite short and focuses on variations of different elements (number of examples per class, etc.) rather than actually add details about the process used by the researchers. Similarly, details about human annotators mentioned in the rebuttal are very sparse (typically you would provide statistics about the annotators, their expertise in CS/other, etc.).

**Reviewer Scores:**

Three reviewers had positive scores and I do not see a reason for them to decrease their scores.  Reviewer qs4b already expressed concerns especially about the few-shot rebuttal results and that they would not raise their score "above a 6" (original rating was 4), implying some increase but a few concerns remaining. Looking at the few-shot results, though, average performance was increased through the proposed method so I do not see a large concern.

Given the original positive scores, likely slightly higher scores post-rebuttal, and extensive rebuttal, I would therefore recommend acceptance. However, I strongly encourage the authors to add rebuttal elements as they do strengthen the paper significantly, and beyond the rebuttal to add significant format example design information ("who did the winnowing", etc.) as well as a discussion of how the proposed method would potentially apply to other scientific domains (a key question for potential readers of the paper).

---

### Decision · Program_Chairs · 2026-01-26

Accept (Poster)